# RNA surveillance via nonsense-mediated mRNA decay is crucial for longevity in *daf-2*/insulin/IGF-1 mutant *C. elegans*

Heehwa G. Son[1,*], Mihwa Seo[1,2,3,*], Seokjin Ham[1], Wooseon Hwang[1], Dongyeop Lee[1], Seon Woo A. An[1], Murat Artan[4], Keunhee Seo[1], Rachel Kaletsky[5], Rachel N. Arey[5], Youngjae Ryu[6], Chang Man Ha[6], Yoon Ki Kim[7,8], Coleen T. Murphy[5], Tae-Young Roh[1,9], Hong Gil Nam[3,10] & Seung-Jae V. Lee[1,2,4]

Long-lived organisms often feature more stringent protein and DNA quality control. However, whether RNA quality control mechanisms, such as nonsense-mediated mRNA decay (NMD), which degrades both abnormal as well as some normal transcripts, have a role in organismal aging remains unexplored. Here we show that NMD mediates longevity in *C. elegans* strains with mutations in *daf-2*/insulin/insulin-like growth factor 1 receptor. We find that *daf-2* mutants display enhanced NMD activity and reduced levels of potentially aberrant transcripts. NMD components, including *smg-2/UPF1*, are required to achieve the longevity of several long-lived mutants, including *daf-2* mutant worms. NMD in the nervous system of the animals is particularly important for RNA quality control to promote longevity. Furthermore, we find that downregulation of *yars-2*/tyrosyl-tRNA synthetase, an NMD target transcript, by *daf-2* mutations contributes to longevity. We propose that NMD-mediated RNA surveillance is a crucial quality control process that contributes to longevity conferred by *daf-2* mutations.

[1] Department of Life Sciences, Pohang University of Science and Technology, Pohang, Gyeongbuk 37673, South Korea. [2] School of Interdisciplinary Bioscience and Bioengineering, Pohang University of Science and Technology, Pohang, Gyeongbuk 37673, South Korea. [3] Center for Plant Aging Research, Institute for Basic Science, Daegu 42988, South Korea. [4] Information Technology Convergence Engineering, Pohang University of Science and Technology, Pohang, Gyeongbuk 37673, South Korea. [5] Department of Molecular Biology & LSI Genomics, Princeton University, Princeton, New Jersey 08544, USA. [6] Research Division, Korea Brain Research Institute, Daegu 41068, South Korea. [7] Creative Research Initiatives Center for Molecular Biology of Translation, Korea University, Seoul 02841, South Korea. [8] Division of Life Sciences, Korea University, Seoul 02841, South Korea. [9] Division of Integrative Biosciences and Biotechnology, Pohang University of Science and Technology, Pohang, Gyeongbuk 37673, South Korea. [10] Department of New Biology, DGIST, Daegu 42988, South Korea. * These authors contributed equally to this work. Correspondence and requests for materials should be addressed to H.G.N. (email: nam@dgist.ac.kr) or to S.-J.V.L. (email: seungjaelee@postech.ac.kr).

A key characteristic of aging is a gradual decline in the quality control of biological system components. Deterioration of DNA and protein quality control plays a central role in aging and age-related diseases. For example, accumulation of DNA damage is linked to age-related neurodegenerative diseases and premature aging, including Werner syndrome[1], while disruption of protein homoeostasis is also closely associated with age-related diseases, including Alzheimer's disease and Parkinson's disease[2]. In addition, mutations of somatic DNA and proteotoxicity caused by the accumulation of misfolded proteins underlie normal aging, and proteostasis is an important component of longevity mechanisms[2,3]. Regarding RNA, several neurodegenerative disorders are associated with defects in RNA-binding protein function[4,5], and many non-coding RNAs, such as microRNAs and long non-coding RNAs, play regulatory roles in longevity[6–8]. In addition, proper RNA splicing is crucial for longevity[9,10]. However, whether RNA quality control affects aging is largely unknown.

Nonsense-mediated mRNA decay (NMD) is a key pathway for maintenance of RNA quality. The NMD complex, which consists of multiple protein components, detects and degrades aberrant transcripts, such as mRNAs containing premature termination codons (PTCs)[11]. NMD also regulates the level of ∼10% of endogenous transcripts, including upstream open reading frames (uORFs)- and long 3′ UTR-containing transcripts[12,13]. Therefore, NMD acts as a crucial regulator of general RNA quality control, and prevents accumulation of potentially deleterious non-functional proteins. The physiological function of NMD is well known in genetic diseases and organismal development[11,14], but it is not yet known whether NMD plays a role in aging processes or in the maintenance of normal function in longevity mutants.

In this report, we show that NMD contributes to longevity conferred by mutations in daf-2, which encodes insulin/insulin-like growth factor 1 (IGF-1) receptor in *C. elegans*. We found that NMD activities declined during aging in various tissues. SMG-2, a key NMD component RNA helicase, was required for the long lifespan of several strains, including daf-2 mutants. RNAi targeting other NMD components, smg-1 through smg-5, also suppressed the longevity of daf-2 mutants. By performing mRNA seq. analysis and RNA half-life measurements, we found that the long-lived daf-2 mutants displayed enhanced NMD activity in a SMG-2-dependent manner. We further showed that down-regulation of an NMD target, yars-2/tyrosyl-tRNA synthetase, contributes to longevity conferred by daf-2 mutations. Together, our data suggest that reduced insulin/IGF-1 signalling increases lifespan through enhancing NMD activity, which is crucial for RNA quality control.

## Results

**NMD activity decreases during aging.** To determine whether NMD-mediated RNA surveillance is important for aging and longevity regulation, we first examined the level of NMD activity using a reporter. This NMD reporter contains a PTC in the first exon of lacZ fused with GFP, sec-23p::gfp::lacZ(PTC) (Fig. 1a)[15]. Therefore, in normal conditions, this transgene is degraded by NMD, resulting in dim GFP. In contrast, when NMD activity is decreased or blocked, GFP intensity is increased (Supplementary Fig. 1a)[15]. We first confirmed that RNAi targeting smg-2/UPF1, a core component of NMD, increased GFP expression (Supplementary Fig. 1b–d). Importantly, aged (Day 9 adult) worms displayed an increased ratio of lacZ(PTC)/lacZ(WT) GFP intensity, indicating an age-dependent impairment of NMD activity (Fig. 1b–d). Next, we examined whether the mRNA levels of PTC-containing rpl-12 (rpl-12(PTC)), a well-established

endogenous target of NMD (Supplementary Fig. 1e)[16], changed during aging. We found that old animals displayed increased rpl-12(PTC) levels when they were normalized with rpl-12(WT) transcripts that do not contain PTC, compared to those in young worms (Fig. 1e). These results support the idea that NMD activity decreases during aging.

We sought to determine in which tissues NMD activity changed during aging by examining neuron-, hypodermis-, muscle- and intestine-specific NMD reporters. We found that the normalized GFP intensities of lacZ(PTC)/lacZ(WT) were increased during aging in the hypodermis, muscle and intestine, while those in neurons were largely unaffected with age (Fig. 1f). This result raises a possibility that the maintenance of NMD activity in neurons is sustained longer than that in other tissues.

**smg-2 is required for the long lifespan of daf-2 mutants.** As we found that NMD activity generally declined during aging, we wondered whether NMD affected organismal lifespan. Loss of smg-2 had a small effect on wild-type (WT) lifespan, suggesting that NMD may not limit normal lifespan. However, smg-2 mutation or RNAi substantially shortened the long lifespan induced by genetic inhibition of the daf-2/insulin/IGF-1 receptor (Fig. 2a–c and Supplementary Fig. 2a,b)[17]. We found that smg-2 mutations also shortened the longevity of mitochondrial respiration-defective isp-1(−) mutants (Fig. 2d), dietary restriction mimetic eat-2(−) mutants (Fig. 2e) and germline-deficient glp-1(−) mutants (Fig. 2f). In contrast to the results with genetic mutants, smg-2 RNAi did not decrease longevity induced by isp-1 (Supplementary Fig. 2c), eat-2 (Supplementary Fig. 2d) or glp-1 (Supplementary Fig. 2e) mutations. smg-2 mutations had no effect on the long lifespan of the hif-1 gain-of-function vhl-1(−) mutants (Fig. 2g) or sensory-defective osm-5(−) mutants (Fig. 2h). Overall, these data indicate that smg-2 is required for lifespan extension rather generally but not universally.

**SMG-2 and DAF-16 may act differently to promote longevity.** DAF-16, a FOXO homologue, is a well-known longevity transcription factor acting downstream of two of the mutants, daf-2(−) and glp-1(−), which we found require smg-2 for their longevity[18,19]. We therefore examined the possibility that smg-2 expression was changed by daf-2 or glp-1 mutations in a daf-16-dependent manner. We found that the mRNA level of smg-2 was not changed by daf-2, glp-1 or daf-16 mutations (Supplementary Fig. 3a,b), or the GFP-fused SMG-2 level was not changed by daf-16 RNAi (Supplementary Fig. 3c). Conversely, we examined whether DAF-16 activity was changed upon inhibition of smg-2. Unexpectedly, we found that smg-2 RNAi increased the nuclear localization of DAF-16::GFP in WT and daf-2(e1368), a weak daf-2 mutant allele amenable for the detection of an increase in nuclear DAF-16 levels (Supplementary Fig. 4a,b). smg-2 mutations also further increased dauer formation, which requires upregulation of DAF-16, in daf-2(e1370) mutants (Supplementary Fig. 4c), and further decreased the lifespan of daf-16; daf-2 double mutants (Supplementary Fig. 4d). These data suggest that SMG-2 and DAF-16 distinctly affect lifespan in daf-2 mutants.

**Many NMD components contribute to longevity in daf-2 mutants.** The NMD pathway is regulated by the NMD complex, composed of many components (SMG-1 through SMG-7). We first determined the efficiency of RNAi clones targeting smg-1 through smg-7. We found that RNAi clones targeting smg-1 through smg-5 decreased the mRNA levels of respective smg genes in WT and daf-2 mutant backgrounds, whereas smg-6 RNAi or smg-7 RNAi

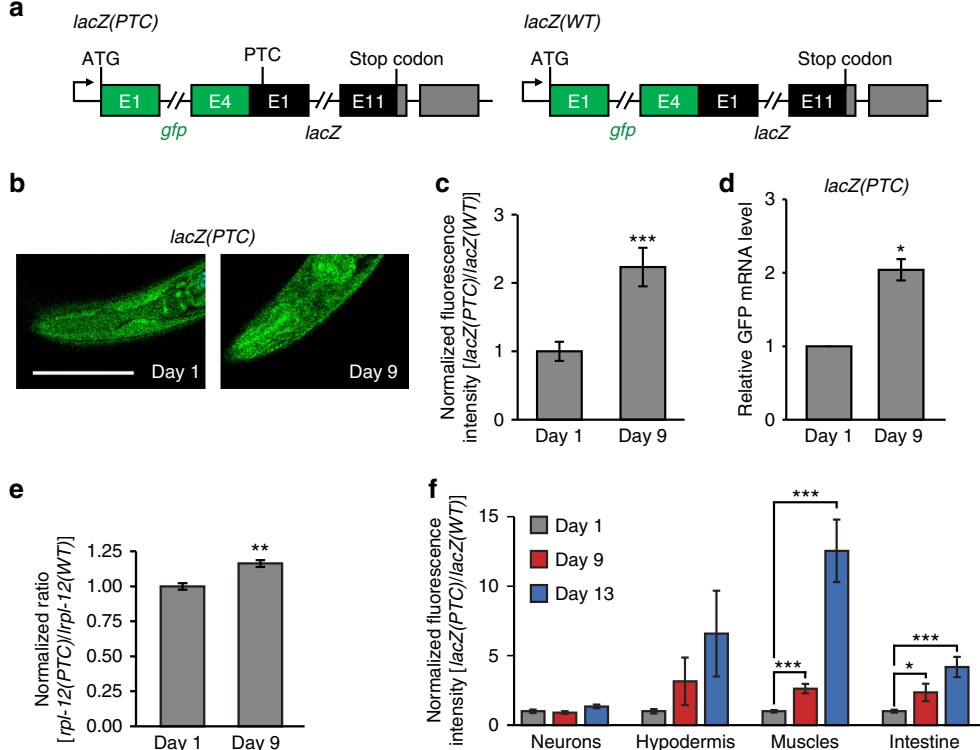

**Figure 1 | NMD activity declines with age.** (**a**) Diagram of a premature termination codon (PTC)-containing GFP-fused NMD reporter, *sec-23p::gfp::lacZ(PTC)* (*lacZ(PTC)*), and a control without PTC, *sec-23p::gfp::lacZ(WT)* (*lacZ(WT)*). Green boxes represent *gfp* exons, black boxes represent *lacZ* exons and grey boxes represent 3′ UTRs. (**b**) The images of young (Day 1) and old (Day 9) NMD reporter, *lacZ(PTC)* (scale bar, 100 μm). (**c**) Normalized fluorescence intensity of *lacZ(PTC)* to age-matched *lacZ(WT)* ($n \geq 23$ from three independent experiments). The same bar graph was also used in Fig. 4i. (**d**) mRNA levels of the GFP-fused NMD reporter, *lacZ(PTC)*, in young (Day 1) and old (Day 9) worms measured by qRT–PCR ($n = 2$). (**e**) Normalized RT–PCR result of PTC-containing *rpl-12* [*rpl-12(PTC)*] by *rpl-12* [*rpl-12(WT)*], which do not contain PTC, transcripts ($n = 3$). (**f**) *lacZ(PTC)* and *lacZ(WT)* were expressed from neuron-, hypodermis-, muscle- and intestinal-specific promoters. Normalized GFP intensities expressed from corresponding tissue-specific promoter-driven *lacZ(PTC)* to age-matched *lacZ(WT)* among different tissues during aging ($n \geq 21$ from three independent experiments). Neuron: *rgef-1p::gfp::lacZ(PTC)/rgef-1p::gfp::lacZ(WT)*, hypodermis: *dpy-7p::gfp::lacZ(PTC)/dpy-7p::gfp::lacZ(WT)*, muscle: *myo-3p::gfp::lacZ(PTC)/myo-3p::gfp::lacZ(WT)* and intestine: *ges-1p::gfp::lacZ(PTC)/ges-1p::gfp::lacZ(WT)*. Hypodermal fluorescence intensity ratio of *lacZ(PTC)/lacZ(WT)* tended to be increased in Day 9 or Day 13 compared to that of Day 1, but the differences were not significant. Error bars represent s.e.m. (two-tailed Student's *t*-test, *$P < 0.05$, **$P < 0.01$, ***$P < 0.001$).

clones did not (Supplementary Fig. 6). We then found that knockdown of *smg-1* through *smg-5* significantly increased the GFP intensity of the NMD reporter (Fig. 3a). The effects of *smg-1* through *smg-5* RNAi on the GFP intensity were variable (Fig. 3a). These data are consistent with reports showing that NMD components play both overlapping and distinct roles, and their effects on a same target can be variable[20]. We found that RNAi knockdown of each of *smg-1*, *-2*, *-3* and *-5* significantly suppressed the long lifespan of *daf-2* mutants (Fig. 3b–d,f); *smg-4* RNAi decreased the long lifespan of *daf-2* mutants in one out of two trials (Fig. 3e). Thus, NMD appears to promote longevity in animals with reduced insulin/IGF-1 signalling.

**NMD is enhanced in *daf-2* mutants.** As NMD plays an important role in RNA surveillance, we examined whether the enhanced NMD underlies the longevity of *daf-2* mutants. To this end, we compared the transcriptomes of WT, *smg-2* mutants, *daf-2* mutants and *smg-2; daf-2* double mutants using mRNA sequencing (Supplementary Fig. 7a). Genes whose expression was changed by *smg-2* mutations significantly overlapped with those by mutations in *smg-1* (ref. 13), which encodes a protein kinase that activates SMG-2 via phosphorylation (Supplementary Fig. 7b); this analysis validates our mRNA sequencing data.

We then analysed NMD target transcripts, such as transcripts with PTCs, transcripts containing uORFs or long 3′ UTRs[11–13,21],

from our mRNA sequencing data. As expected, many PTC-containing transcripts, identified by using SpliceR[22], were highly expressed by *smg-2* mutations (Supplementary Fig. 8a). Transcripts that were highly expressed in *smg-2(−)* compared to those in WT had a higher tendency to have uORFs (Supplementary Fig. 8c). In addition, the levels of transcripts with long 3′ UTRs tended to be increased by *smg-2(−)* (Supplementary Fig. 8e). Importantly, we found that *daf-2* mutations reduced the expression of PTC-, uORF- or long 3′ UTR-containing mRNAs (Fig. 4a–c and Supplementary Fig. 9). Next, we examined whether the expression of PTC-, uORF- or long 3′ UTR-containing transcripts that were downregulated in *daf-2* mutants compared to WT animals were changed in *smg-2; daf-2* mutants. The expression levels of downregulated PTC-, uORF- or 3′ UTR-containing transcripts in *daf-2* mutants were increased in *smg-2(−); daf-2(−)* mutants (Fig. 4d–f). We then confirmed our RNA-sequencing data for PTC-containing transcripts using qRT–PCR, the NMD reporter and the RNA half-life assay. We found that the mRNA level of PTC-containing *rpl-12* measured by using qRT–PCR was decreased in *daf-2* mutants, in a *smg-2*-dependent manner (Fig. 4g,h). In addition, *daf-2* mutations decreased the fluorescence intensity ratio of *lacZ(PTC)/lacZ(WT)* GFP reporters, both in young and old animals (Fig. 4i). We also found that RNA half-lives of selected NMD targets were reduced by *daf-2* mutations (Fig. 4j,k).

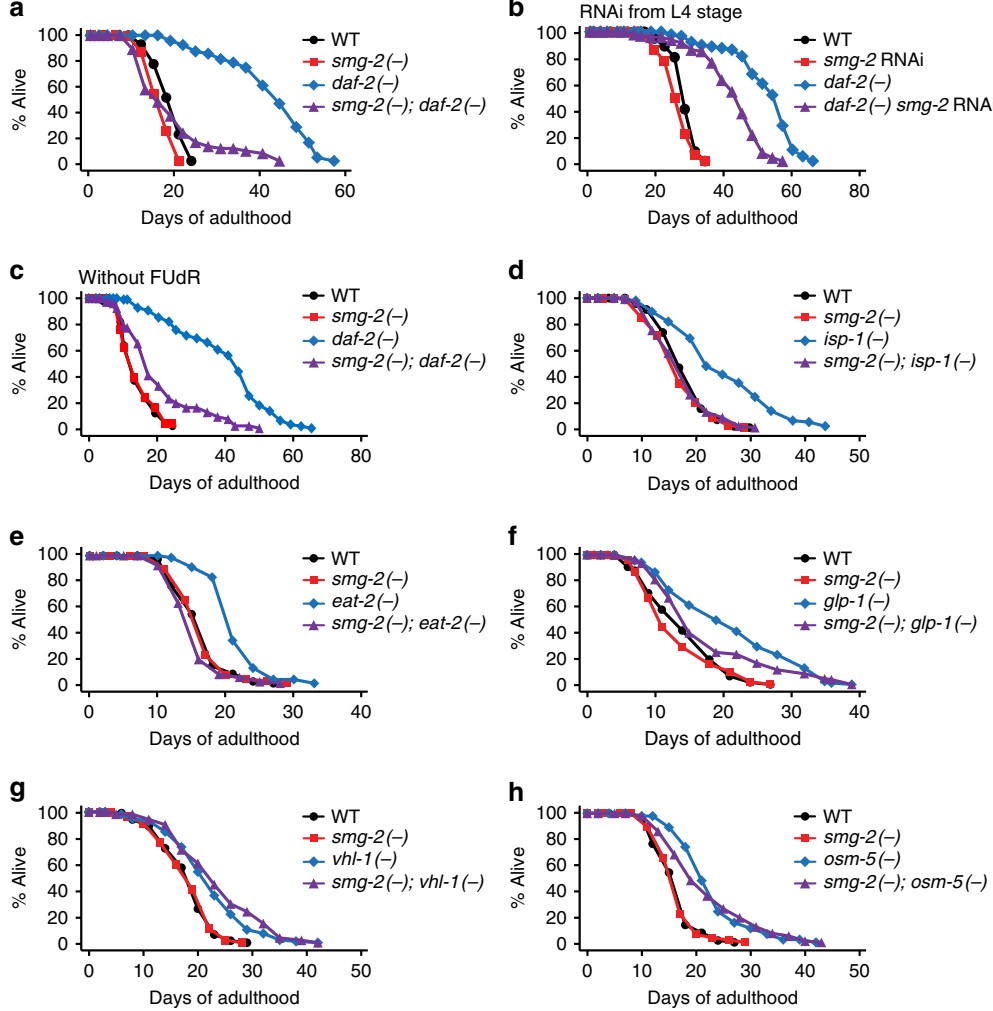

**Figure 2 | NMD is required for the long lifespan of insulin/IGF-1 receptor *daf-2* mutants.** (**a**) The long lifespan of *daf-2(e1370)* (*daf-2(−)*) mutants was significantly suppressed by *smg-2(qd101)* (*smg-2(−)*) mutations. (**b**) *smg-2* RNAi from L4 stage significantly decreased the longevity of *daf-2(−)* animals. (**c**) *smg-2* mutations significantly reduced the long lifespan conferred by *daf-2* mutations without FUdR treatment. (**d–f**) Longevity induced by *isp-1(qm150)* (*isp-1(−)*) (**d**), *eat-2(ad1116)* (*eat-2(−)*) (**e**) or *glp-1(e2141)* (*glp-1(−)*) mutations (**f**) was suppressed by *smg-2* mutations. (**g,h**) *smg-2* mutations had no effect on the long lifespan of *vhl-1(ok161)* (*vhl-1(−)*) (**g**) or *osm-5(p813)* (*osm-5(−)*) (**h**) mutants. Please see Supplementary Fig. 2c–f for data regarding the effects of *smg-2* RNAi on the longevity of *isp-1*, *eat-2*, *glp-1* and *vhl-1* mutants. Specific lifespan-decreasing effects of *smg-2* RNAi on *daf-2* mutants may be due to variability in RNAi efficiency, as we did not perform all the RNAi-based lifespan assays using various longevity mutants in the same experimental set. *smg-2* mutations also partially suppressed delayed age-dependent declines in the body movement of *daf-2* mutants (Supplementary Fig. 5). See Supplementary Data 1 for statistical analysis and additional repeats.

Moreover, the effect of *daf-2* mutations on RNA half-lives was dependent on *smg-2* (Fig. 4j,k). Overall, these data suggest that NMD is enhanced in *daf-2* mutants.

**smg-1 overexpression increases lifespan**. Next, we generated transgenic animals expressing GFP-fused SMG-2, driven by its own promoter (*smg-2p::smg-2::gfp*) (Fig. 5a). SMG-2::GFP was expressed in the cytoplasm of cells, consistent with the finding that NMD functions in the cytoplasm (Fig. 5a)[23]. Tissues that expressed the *smg-2::gfp* transgene included neurons, intestine and muscles (Fig. 5a). The *smg-2::gfp* transgene was functional, as it rescued the lifespan phenotypes in *smg-2; daf-2* mutants and in *smg-2* mutants (Fig. 5b,c). *smg-2* overexpression did not affect the lifespan of WT animals (Fig. 5d). In contrast, overexpression of SMG-1, a protein kinase that phosphorylates and activates SMG-2, increased lifespan (Fig. 5e). Together with the finding that *smg-2* mRNA level was not changed by *daf-2*, *glp-1* or *daf-16* mutations (Supplementary Fig. 3a,b), these

data indicate that post-transcriptional regulation of SMG-2 via phosphorylation by SMG-1 may exert its lifespan-extending effects.

**Neuronal NMD plays a major role in lifespan extension**. We then examined in which tissues *smg-2* contributed to the long lifespan of *daf-2* mutants. We found that neuron-specific *smg-2* RNAi largely suppressed the long lifespan of *daf-2* mutants (Fig. 6a); in contrast, hypodermis-, muscle- or intestine-specific *smg-2* RNAi had little to no effect on longevity (Fig. 6b–d and Supplementary Fig. 11). Using tissue-specific rescue experiments, we found that neuron-, hypodermis- and muscle-specific *smg-2* expression partially restored longevity in *smg-2(−); daf-2(−)* animals (Fig. 6e–g), while intestine-specific *smg-2* expression did not (Fig. 6h). Thus, neurons are the only tissue that is required and sufficient for longevity in *daf-2(−)* mutants. In combination with the requirement of neuron-specific *smg-2* activity and the neuronal maintenance of NMD, these data suggest that RNA

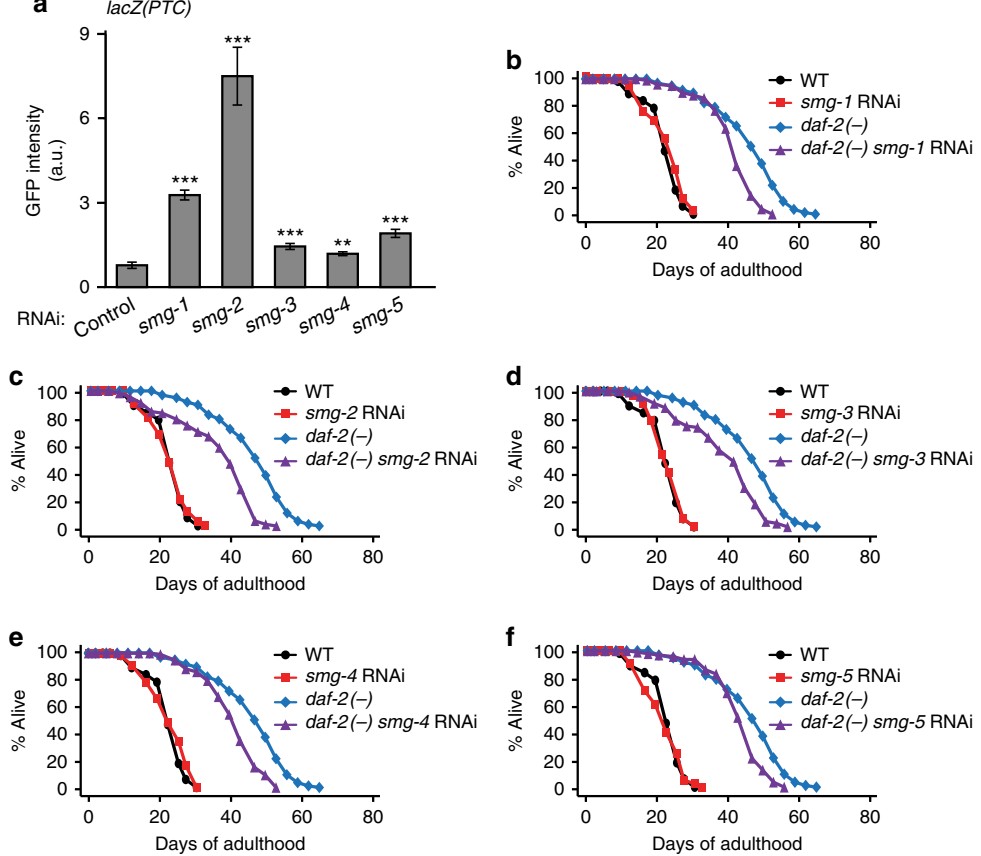

**Figure 3 | Several NMD components are required for the full longevity of *daf-2* mutants.** (**a**) Quantified GFP fluorescence intensity of an NMD reporter, *sec-23p::gfp::lacZ(PTC)* (*lacZ(PTC)*), upon knockdown of *smg* genes (*n* = 16 from two independent experiments). Error bars represent s.e.m. (two-tailed Student's *t*-test, **P < 0.01, ***P < 0.001). (**b–f**) RNAi knockdown of each NMD component, *smg-1* (**b**), *smg-2* (**c**), *smg-3* (**d**), *smg-4* (one out of two trials) (**e**) or *smg-5* (**f**), shortened the longevity of *daf-2(e1370)* (*daf-2( − )*) mutants. See Supplementary Data 1 for statistical analysis and additional repeats.

surveillance in neurons is particularly crucial for the longevity of *daf-2* mutants.

We then analysed transcripts expressed in neurons of *daf-2* mutants and WT animals[24], and found that overall expression of PTC-, uORF- or long 3′ UTR-containing transcripts was significantly decreased in the neurons of *daf-2* mutants compared to those of WT (Fig. 6i–k). In addition, we noticed that genes repressed in *daf-2* mutants in a *smg-2*-dependent manner were enriched in neuronal gene ontology (GO) terms, such as axon guidance (Supplementary Fig. 12a). These data are consistent with the idea that neurons are a crucial tissue for NMD to maintain the long lifespan of animals with reduced insulin/IGF-1 signalling (IIS).

**Decreased *yars-2* levels contribute to longevity**. Next, we sought to identify targets of NMD that mediate the long lifespan of *daf-2* mutants. Because the main role of NMD is degradation of its target mRNAs, we focused on mRNAs that were downregulated in *daf-2* mutants. We performed GO analysis with PTC- or uORF-containing transcripts, which are potential NMD targets, and whose levels were decreased in *daf-2* mutants in a *smg-2*-dependent manner. We found that a GO term regarding tyrosyl-tRNA aminoacylation, which includes *yars-1* and *yars-2*, was highly enriched (Fig. 7a and Supplementary Fig. 12c); *yars-1* and *yars-2* encode cytosolic and mitochondrial tyrosyl-tRNA synthetases, respectively. We confirmed that mRNA levels of *yars-2b.1*, one of the isoforms of *yars-2* and an NMD target, were significantly decreased in *daf-2* mutants in a *smg-2*-dependent manner (Fig. 7b and Supplementary Fig. 13a,b). We then tested

whether changes in *yars-2* mRNA levels by *smg-2* contributed to the longevity of *daf-2* mutants, by performing lifespan assays upon knocking down *yars-2*. We found that *yars-2* RNAi restored longevity in *smg-2*; *daf-2* mutants (Fig. 7c), while having no effect on the lifespan of WT, *smg-2* or *daf-2* mutants (Supplementary Fig. 13c,d). These data suggest that an NMD-dependent decrease in *yars-2*/tRNA synthetase mRNA levels is crucial for the longevity of *daf-2* mutants.

**Discussion**

NMD is an evolutionarily conserved pathway crucial for RNA surveillance. Here our data suggest that NMD-mediated RNA quality control is critical for longevity in *C. elegans*. Interestingly, neurons are at least partially resistant to age-dependent declines in NMD activity, and NMD in neurons contributes to longevity conferred by *daf-2*/insulin/IGF-1 receptor mutations. Neurons in the mammalian central nervous system are mostly post-mitotic and must function for a longer time than other cells without diluting toxic RNA through cell division. Thus, neurons may need to maintain better RNA quality control systems via enhanced NMD.

One interesting aspect of our study is that the efficiency of NMD in different tissues during the course of aging appears to be different. NMD efficiencies also differ among various mammalian tissues[25]. For example, the ratio of PTC-containing *Men1* mRNA to control is variable among 13 different mouse tissues[26]. One possible explanation regarding these tissue-specific effects of NMD is that NMD components are differentially regulated among various tissues by cell type-specific negative feedback

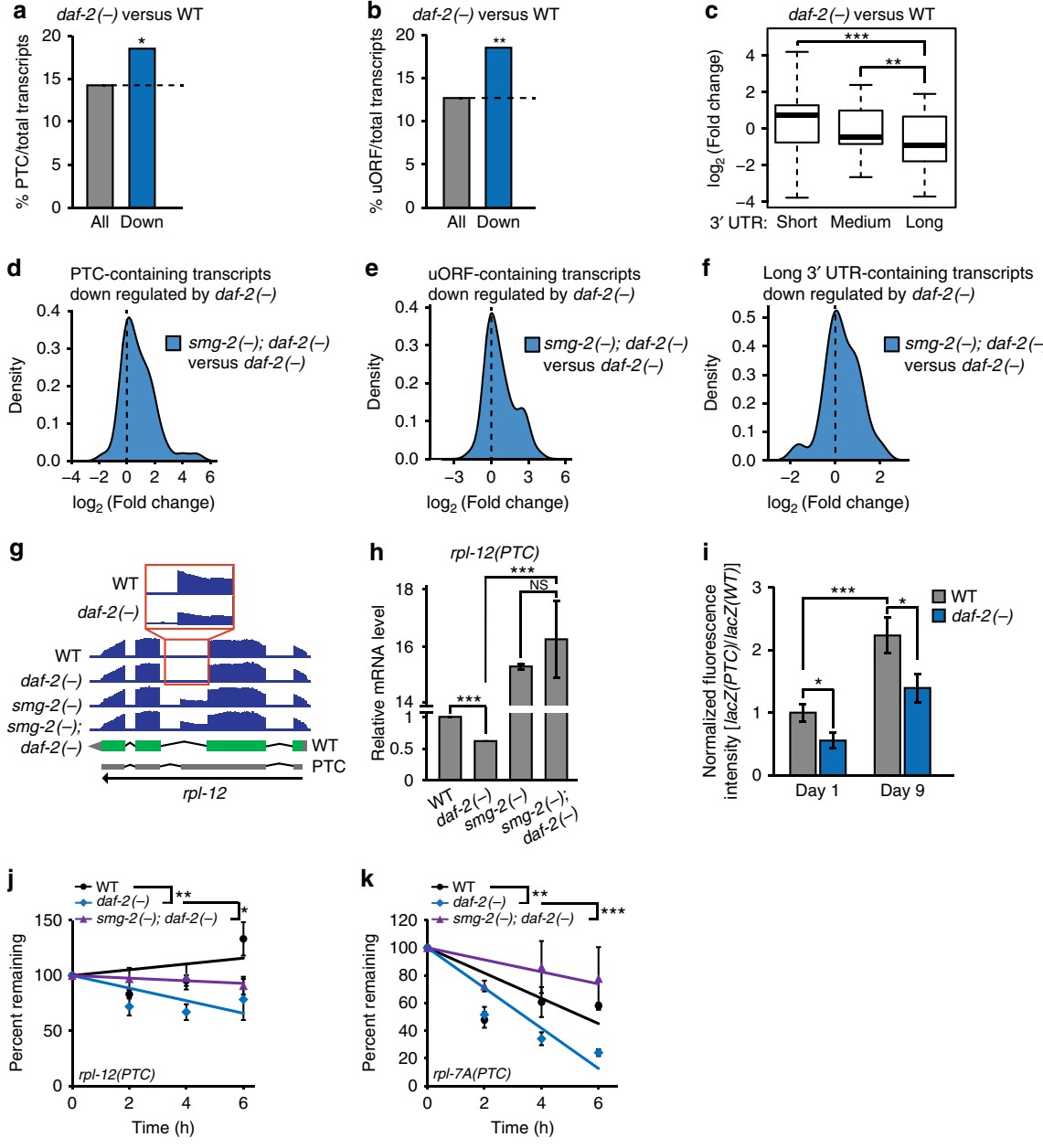

**Figure 4 | Reduced IIS decreases the levels of NMD targets.** (**a,b**) The fractions of PTC- (**a**) or uORF- (**b**) containing transcripts per total transcripts in *daf-2(e1370)* (*daf-2(−)*) compared to WT (Down: log$_2$(fold change) ≤ −1, $P ≤ 0.1$, *$P < 0.05$, **$P < 0.01$ $\chi^2$-test). See Supplementary Table 1 for actual $P$ values, and Supplementary Data 3 for lists of PTC- and uORF-containing transcripts. (**c**) Box plots showing the fold changes in *daf-2(−)* mutants compared to WT for the transcripts containing long, medium or short 3′ UTR lengths. Short (≤350 nt), medium (350 nt < < 1,500 nt) and long (≥1,500 nt). 3′ UTR lengths were categorized following a previous report[50]. Thick black lines indicate median values. Bottom and top of the box plot represent 25th and 75th percentile, respectively, and whiskers indicate the data within the 1.5 interquartile range, which is the distance between the lower and upper quartiles of the data (**$P < 0.01$, ***$P < 0.001$, Wilcox rank-sum test). (**d–f**) *smg-2* dependency analysis of the PTC- (**d**), uORF- (**e**) or long 3′ UTR- (**f**) containing transcripts whose levels were decreased in *daf-2(−)* mutants compared to WT (Down: log$_2$(fold change) ≤ −1, $P ≤ 0.1$). We also analysed NMD targets (uORF-, PTC- and long 3′ UTR-containing transcripts) using *smg-1* mutant data[13] (Supplementary Fig. 8b,d,f). (**g,h**) The effects of *daf-2(−)* and *smg-2(−)* mutations on the expression level of *rpl-12(PTC)*, an endogenous *C. elegans* NMD target[16], measured by using mRNA sequencing (**g**) and qRT–PCR ($n = 2$) (**h**). Green rectangles indicate exons, and grey parts indicate untranslated regions. (**i**) The fluorescence intensity of *lacZ(PTC)* was normalized to age- and genotype-matched *lacZ(WT)*. ($n ≥ 16$ from at least two independent experiments). The same bar graphs for WT control was used in Fig. 1c (two-tailed Student's *t*-test, *$P < 0.05$, ***$P < 0.001$). (**j,k**) The mRNA half-lives of *C. elegans* NMD targets, *rpl-12(PTC)* (**j**) and *rpl-7A(PTC)* (**k**), in adult worms ($n = 3$). $P$ values were calculated by using two-way ANOVA test (*$P < 0.05$, **$P < 0.01$, ***$P < 0.001$). See Supplementary Fig. 10 for the optimization of RNA half-life assays in *C. elegans*. Error bars represent s.e.m.

regulation[27]. Further studies are required to dissect mechanisms by which NMD is tissue-specifically regulated.

Our data suggest that enhanced NMD increases lifespan in *daf-2* mutants by efficiently degrading *yars-2b.1*/tyrosyl-tRNA synthetase isoform b.1. Genetic inhibition of tRNA synthetases, which may decrease mRNA translation, extends *C. elegans* lifespan[28,29]. *daf-2* mutants display reduced translation rates[30], and reduced translation confers longevity in multiple organisms[31].

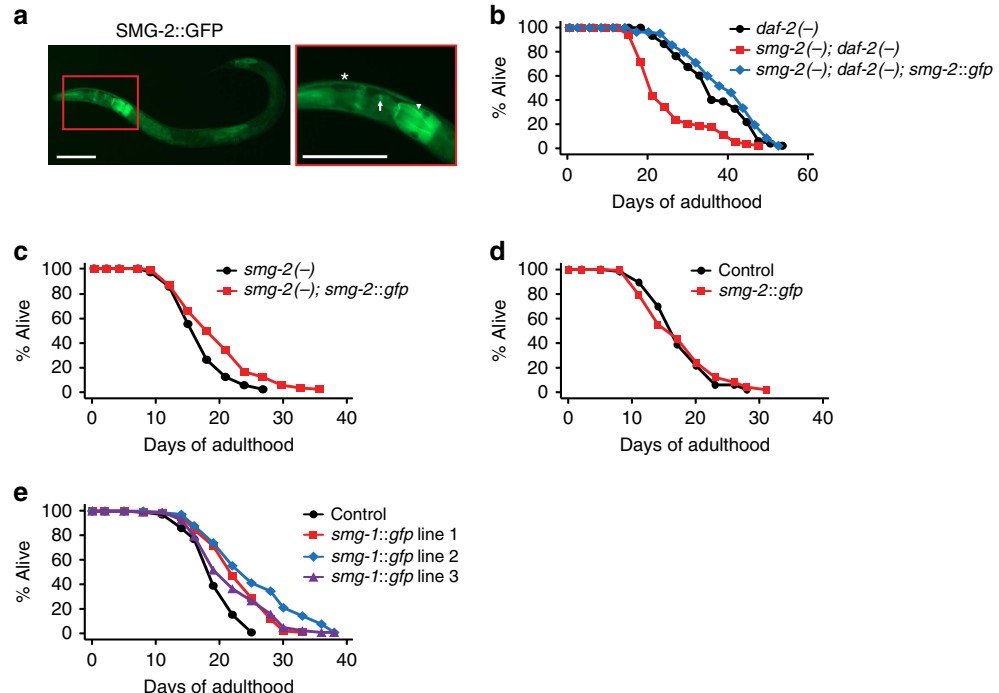

**Figure 5 | smg-1 overexpression increases lifespan.** (**a**) Images of SMG-2::GFP-expressing worms (scale bar, 100 µm) (arrow: neurons, arrowhead: intestine, asterisk: muscles). (**b,c**) The smg-2::gfp transgene restored longevity in smg-2( − ); daf-2( − ) mutants (**b**), and in smg-2( − ) animals (**c**). (**d**) The smg-2::gfp transgene had no significant effect on the lifespan of control (odr-1p::rfp) worms. (**e**) Three independent smg-1 transgenic lines increased the lifespan of control (odr-1p::rfp) worms. smg-1::gfp line 1 increased lifespan in one out of three trials. See Supplementary Data 1 for statistical analysis and additional repeats.

Thus, increased NMD activity by reduced IIS may prevent age-dependent increases in abnormal transcripts, including tRNA synthetase genes, and this may in turn contribute to the longevity of daf-2 mutants by decreasing translation. It is intriguing that reduced IIS may promote healthy aging by increasing the quality of RNA and proteins by affecting both transcription and translation.

Emerging evidence suggests that RNA toxicity underlies various degenerative diseases, yet its role in aging or longevity regulation has not been previously demonstrated. We report that IIS, one of the most evolutionarily conserved aging-regulatory signalling pathways, regulates RNA surveillance through NMD to influence aging. It is well established that long-lived daf-2 mutants display enhanced protein homoeostasis and genome maintenance[2,32,33]. Our findings suggest that IIS actively regulates quality control systems for all three components of the Central Dogma (DNA, RNA and proteins), and this may at least in part underlie the extreme longevity of daf-2 mutants. Furthermore, our mutant studies indicate that NMD may be an important shared component of several longevity pathways, including dietary restriction, germline loss and mitochondrial mutations. As humans live much longer than animals with comparable sizes and metabolic rates, and NMD components are well conserved, it is likely that humans are equipped with systems that maintain cellular RNA homoeostasis as well.

## Methods

**Strains.** All strains were maintained at 20 °C. Some strains were provided by the Caenorhabditis Genetics Center, which is funded by NIH Office of Research Infrastructure Programs (P40 OD010440). The following strains were used in this study. N2 WT, IJ445 smg-2(qd101) I outcrossed four times with N2, CF1041 daf-2(e1370) III outcrossed six times to N2, IJ446 smg-2(qd101) I; daf-2(e1370) III, IJ7 vhl-1(ok161) X outcrossed three times to N2, IJ1280 smg-2(qd101) I; vhl-1(ok161) X, IJ173 eat-2(ad1116) II outcrossed four times to N2, IJ1239 smg-2(qd101) I; eat-2(ad1116) II, CF1903 glp-1(e2141) III, IJ1238 smg-2(qd101) I; glp-1(e2141) III, CF2172 isp-1(qm150) IV, IJ1281 smg-2(qd101) I; isp-1(qm150) IV, CF2553 osm-5(p813) X, IJ1285, smg-2(qd101) I; osm-5(p813) X, WM27

rde-1(ne219) V, NR222 rde-1(ne219) V; kzIs9[lin-26p::nls::gfp; lin-26p::rde-1; rol-6D], NR350 rde-1(ne219) V; kzIs20[hlh-1p::rde-1; sur-5p::nls::gfp], VP303 rde-1(ne219) V; kbIs7[nhx-2p::rde-1; rol-6D], JM43 rde-1(ne219) V; Is[wrt-2p::rde-1; myo-2p::rfp], IJ411 daf-2(e1370) III; rde-1(ne219) V, IJ415 daf-2(e1370) III; rde-1(ne219) V; kzIs9[lin-26p::nls::gfp; lin-26p::rde-1; rol-6D], IJ416 daf-2(e1370) III; rde-1(ne219) V; kzIs20[hlh-1p::rde-1; sur-5p::nls::gfp], IJ417 daf-2(e1370) III; rde-1(ne219) V; kbIs7[nhx-2p::rde-1; rol-6D], IJ418 daf-2(e1370) III; rde-1(ne219) V; Is[wrt-2p::rde-1; myo-2p::rfp], NL3321 sid-1(pk3321) V, TU3401 sid-1(pk3321) V; uIs69[myo-2p::mCherry; unc-119p::sid-1], IJ807 daf-2(e1370) III; sid-1(pk3321) V, IJ898 daf-2(e1370) III; sid-1(pk3321) V; uIs69[myo-2p::mCherry; unc-119p::sid-1], [sec-23p::gfp::lacZ(PTC)], [sec-23p::gfp::lacZ], IJ344 daf-2(e1370) III; [sec-23p::lacZ(PTC)], IJ390 daf-2(e1370) III; [sec-23p::gfp::lacZ], IJ1097 yhEx247[smg-2p::smg-2::gfp, odr-1p::rfp], IJ1100 smg-2(qd101) I; yhEx247[smg-2p:: smg-2:: gfp, odr-1p::rfp], IJ1093 yhEx246[odr-1p::rfp], IJ1094 smg-2(qd101) I; yhEx246[odr-1p::rfp], IJ1095 daf-2(e1370) III; yhEx246[odr-1p::rfp], IJ1096 smg-2(qd101) I; daf-2(e1370) III; yhEx246[odr-1p::rfp], IJ1106 smg-2(qd101) I; daf-2(e1370) III; yhEx247[smg-2p::smg-2::gfp; odr-1p::rfp], IJ1332 yhEx365[dyp-7p:: gfp::lacZ(PTC); rol-6D], IJ1333 yhEx366[ges-1p::gfp::lacZ(PTC); rol-6D], IJ1334 yhEx353[myo-3p::gfp::lacZ(PTC); rol-6D], IJ1335 yhEx354[rgef-1p::gfp::lacZ(PTC); rol-6D], IJ1336 yhEx355[dyp-7p::gfp::lacZ; rol-6D], IJ1337 yhEx356[ges-1p::gfp::lacZ; rol-6D], IJ1338 yhEx357[myo-3p::gfp::lacZ; rol-6D], IJ1339 yhEx358[rgef-1p:: gfp::lacZ; rol-6D], CF1290 N2 carrying pRF4(rol-6(su1006)) alone, IJ1227 yhEx329[smg-1p::smg-1::gfp; odr-1p::rfp], IJ1228 yhEx330[smg-1p::smg-1::gfp; odr-1p::rfp], IJ1306 yhEx331[smg-1p::smg-1::gfp; odr-1p::rfp], IJ1301 smg-2(qd101) I; daf-2(e1370) III; yhEx323[ges-1p::smg-2::gfp; odr-1p::rfp], IJ1302 smg-2(qd101) I; daf-2(e1370) III; yhEx324[rgef-1p::smg-2::gfp; odr-1p::rfp], IJ1303 smg-2(qd101) I; daf-2(e1370) III; yhEx325[hlh-1p::smg-2::gfp; odr-1p::rfp], IJ1304 smg-2(qd101) I; daf-2(e1370) III; yhEx326[dpy-7p::smg-2::gfp; odr-1p::rfp], IJ382 daf-16(mu86) I, CF1085 daf-16(mu86) I; daf-2(e1370)III, CF1880 daf-16(mu86) I; glp-1(e2141) III, IJ1058 daf-16(mu86) I; muIs112[daf-16p::GFP::daf-16cDNA; odr-1p::rfp], CF2688 daf-16(mu86) I; daf-2(e1368) III; muIs112[daf-16p::GFP::daf-16cDNA; odr-1p::rfp].

**Lifespan assays.** Lifespan assays were performed as described previously with some modifications[34]. For mutant lifespan assays, OP50 bacteria were cultured in Luria broth (LB) containing 10 µg ml$^{-1}$ streptomycin (Sigma, St Louis, MO, USA) overnight at 37 °C. The overnight OP50 culture (100 µl) was then seeded on nematode growth media (NGM) plates and incubated overnight at 37 °C. To prevent progeny from hatching, 5-fluoro-2′-deoxyuridine (FUdR, Sigma, St Louis, MO, USA) was added to the plates at a final concentration of 5 µM for most lifespan assays. Day 1 adult worms were placed on fresh plates, and then transferred to new plates after 1 or 2 days. For RNAi-mediated lifespan assays, each

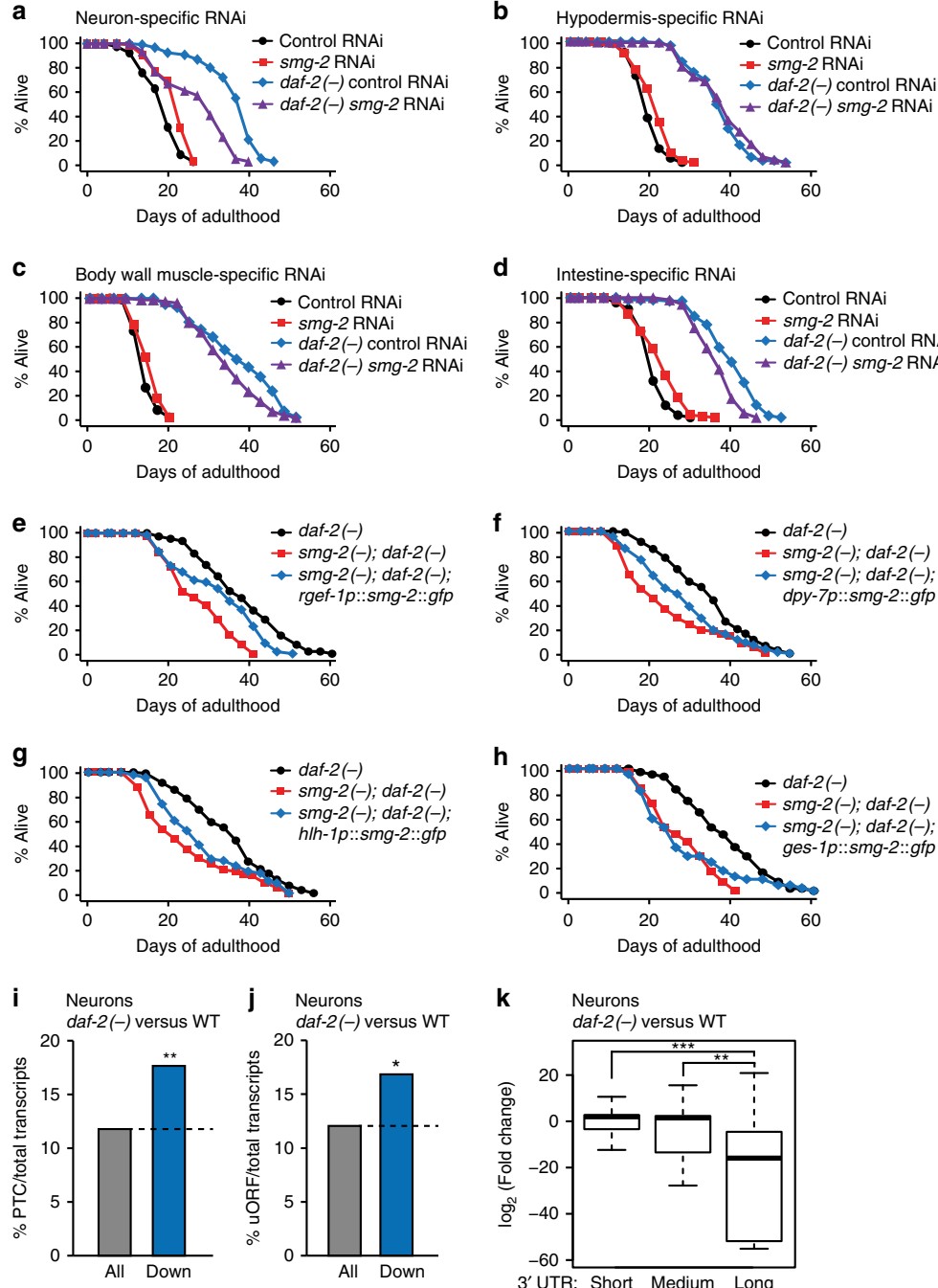

**Figure 6 | Neuronal *smg-2* plays a major role in the long lifespan of *daf-2* mutants.** (**a–d**) Effects of neuron- (**a**), hypodermis- (**b**), body wall muscle- (**c**) and intestine- (**d**) specific RNAi knockdown of *smg-2* on the longevity of *daf-2*( − ) animals. See Supplementary Fig. 11 for the lifespan graphs of control and another hypodermis-specific RNAi strain. (**e–h**) The lifespan curves with transgenic animals expressing *smg-2::gfp* in neurons (**e**), the hypodermis (**f**), muscles (**g**) and the intestine (**h**) in *smg-2(qd101); daf-2(e1370)* (*smg-2*( − ); *daf-2*( − )) mutants. See Supplementary Data 1 for statistical analysis and additional repeats. (**i,j**) Analysis of PTC- (**i**) or uORF- (**j**) containing transcripts among transcripts downregulated (Down: $\log_2$(fold change) $\leq -1$, $P \leq 0.1$) in the neurons of *daf-2*( − ) mutants compared to those of WT[24] (*$P < 0.05$, **$P < 0.01$, $\chi^2$-test). See Supplementary Table 1 for actual $P$ values. (**k**) Box plots showing fold changes of neuronal transcripts in *daf-2* mutants compared to those in WT, categorized by different 3′ UTR lengths. Long ($\geq 1,500$ nt), medium ($350$ nt$<<1,500$ nt) and short ($\leq 350$ nt). 3′ UTR lengths were defined following a previous report[50]. Thick black lines indicate median values. Bottom and top of the box plot represent 25th and 75th percentile, respectively, and whiskers indicate the data within the 1.5 interquartile range, which is the distance between the lower and upper quartiles of the data (**$P < 0.01$, ***$P < 0.001$, Wilcox rank-sum test).

RNAi clone-expressing HT115 bacteria was cultured in LB containing 50 µg ml$^{-1}$ ampicillin (USB, Santa Clara, CA, USA) overnight at 37 °C. The cultured HT115 bacteria in LB were seeded onto NGM plates and incubated overnight at 37 °C. One millimolar isopropylthiogalactoside (IPTG, Gold biotechnology, St Louis, MO, USA) was added and incubated at room temperature for ∼24 h. The plates were then treated with FUdR at a final concentration of 5 µM. Age-synchronized Day 1 adult worms were placed on each RNAi clone-expressing HT115-seeded plate, and were transferred to new plates after 1 or 2 days. All lifespan assays were performed at 20 °C, except for temperature-sensitive *glp-1(e2141)* mutants that display germline deficiency at 25 °C. The *glp-1* mutants were allowed to develop to adults at 25 °C, and age-synchronized Day 1 adult worms were transferred to replete plates without FUdR at 20 °C. In the case of lifespan assays conducted without

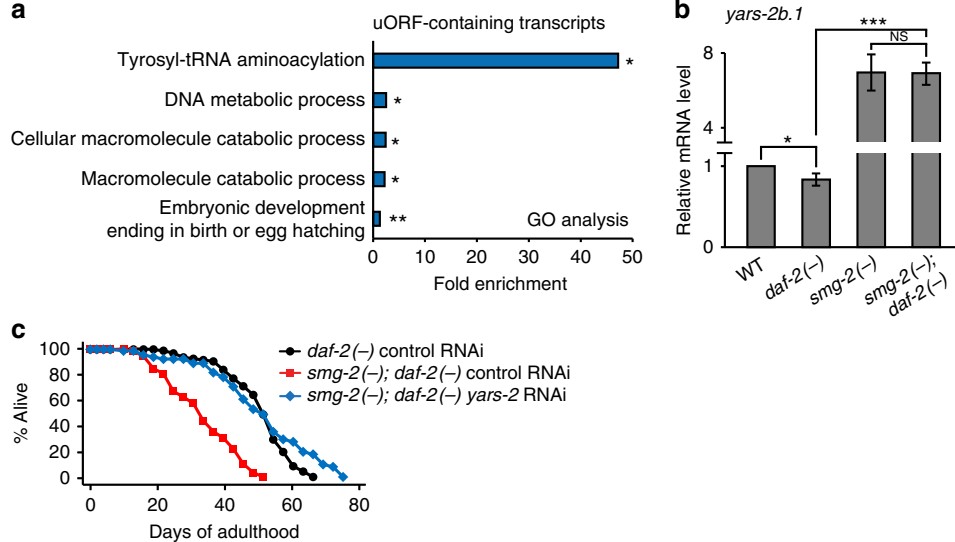

**Figure 7 | Downregulation of *yars-2* promotes longevity in *daf-2* mutants.** (**a**) GO terms that were enriched among uORF-containing transcripts whose levels were decreased by *daf-2* mutations in a *smg-2*-dependent manner (log$_2$(fold change) ≤ − 0.4, *$P$ < 0.05, **$P$ < 0.01, modified Fisher's exact tests[51]). See Supplementary Data 2 for the lists of transcripts that were used for GO analysis. See Supplementary Fig. 12b for GO terms enriched among genes that were upregulated in *daf-2* mutants compared to WT animals. (**b**) mRNA levels of *yars-2b.1*, an NMD target, were measured by qRT–PCR ($n \geq 3$). Error bars represent s.e.m. (two-tailed Student's *t*-test, *$P$ < 0.05, ***$P$ < 0.001). WT, *daf-2(e1370)* (*daf-2(−)*), *smg-2(qd101)* (*smg-2(−)*) and *smg-2(qd101); daf-2(e1370)* (*smg-2(−); daf-2(−)*) were used. The mRNA levels of *yars-2a* isoform, which is not a target of NMD, were not significantly changed by *daf-2* mutations (Supplementary Fig. 13e,f). The mRNA levels of *yars-1* were not changed in our qRT-PCR data (Supplementary Fig. 13g). The mRNA levels of *yars-2b.1* were decreased in long-lived *isp-1* and *glp-1* mutants (Supplementary Fig. 13h,i). (**c**) Effects of *yars-2* RNAi on the lifespan of *smg-2(−); daf-2(−)*. See Supplementary Data 1 for statistical analysis and additional repeats.

FUdR, worms were transferred to new plates every 2–3 days until they ceased to lay eggs. Worms that did not respond to a gentle touch using a platinum wire were counted as dead worms. Worms that crawled off the plates, ruptured, bagged or burrowed were censored but included in the statistical analysis. Statistical analysis of lifespan results was processed by using OASIS (online application of survival analysis, http://sbi.postech.ac.kr/oasis)[35], and *P* values were calculated by using log-rank (Mantel–Cox method) test. For tissue-specific RNAi lifespan assay, following mutants were used (Neurons: *sid-1(pk3321); uIs69[myo-2p::mCherry; unc-119p::sid-1]* and *daf-2(e1370); sid-1(pk3321); uIs69[myo-2p::mCherry; unc-119p::sid-1]*. Intestine: *rde-1(ne219); kbIs7[nhx-2p::rde-1; rol-6D]* and *daf-2(e1370); rde-1(ne219); kbIs7[nhx-2p::rde-1; rol-6D]*. Body wall muscles: *rde-1(ne219); kzIs20[hlh-1p::rde-1; sur-5p::nls::gfp]* and *daf-2(e1370); rde-1(ne219); kzIs20[hlh-1p::rde-1; sur-5p::nls::gfp]*. Hypodermis: *rde-1(ne219); kzIs9[lin-26p::nls::gfp; lin-26p::rde-1; rol-6D], daf-2(e1370); rde-1(ne219); kzIs9[lin-26p::nls::gfp; lin-26p::rde-1; rol-6D], rde-1(ne219); Is[wrt-2p::rde-1; myo-2p::rfp]*, and *daf-2(e1370); rde-1(ne219); Is[wrt-2p::rde-1; myo-2p::rfp]*.).

**Cloning and generation of transgenic worms.** *smg-2::smg-2::gfp* transgenic animals were generated as described previously with some modifications[17]. The coding region of *smg-2* (∼3.2 kb) was amplified by using PCR fusion methods[36]. The PCR products of a promoter (∼0.7 kb upstream of the start codon) and the coding region of *smg-2* were cloned into pDEST R4-R3 vector by using the Gateway cloning system (Invitrogen, Carlsbad, CA, USA). A mixture of *smg-2p::smg-2::gfp* (25 ng µl$^{-1}$) and *odr-1p::rfp* (75 ng µl$^{-1}$) was injected into Day 1 adult N2 worms. To generate tissue-specific promoter-driven *smg-2::gfp* constructs, pPD95.75 (Fire lab *C. elegans* vector kit) was cut by Acc65I, and then PCR-amplified coding region of *smg-2* (∼3.2 kb) was inserted into the linearized vector using In-fusion HD cloning kit (Takara). The vector was digested by same enzyme Acc65I, and then promoter regions of *ges-1* (∼2 kb), *rgef-1* (∼2 kb), *dpy-7* (∼0.5 kb) or *hlh-1* (∼2 kb) were inserted into the linearized *smg-2::gfp* vector to generate intestine-, neuron-, hypodermis- or muscle-specific *smg-2::gfp* constructs, respectively. To generate transgenic worms, each tissue-specific construct (25 ng µl$^{-1}$) was injected with *odr-1p::rfp* (75 ng µl$^{-1}$) into Day 1 adults. To generate *smg-1p::smg-1::gfp*, a *smg-1* promoter region (∼1.5 kb) was amplified by using PCR, and then inserted into linearized pPD95.75 (Acc65I cut) by using In-fusion HD cloning kit. The *smg-1p::gfp* vector was digested by Acc65I, and the coding region of *smg-1* isoform a (∼7 kb), which was amplified by using PCR, was inserted into the linearized vector by using In-fusion HD cloning kit. To generate *smg-1p::smg-1::gfp* transgenic worms, the mixture of *smg-1p::smg-1::gfp* (25 ng µl$^{-1}$) and *odr-1p::rfp* (75 ng µl$^{-1}$) were injected into Day 1 adult worms. To generate tissue-specific NMD reporters and controls, *gfp::lacZ*-containing pPD96.04 (Fire lab *C. elegans* vector kit) was used. Through site-directed mutagenesis, C to T mutation was introduced to create stop codon on the first exon

of *lacZ* as described previously[15]. The original pPD96.04 and mutagenized pPD96.04 vectors were cut by Acc65I, and promoter regions of *ges-1* (∼2 kb), *rgef-1* (∼2 kb), *dpy-7* (∼0.5 kb) or *myo-3* (2 kb) were inserted into the original pPD96.04 and mutagenized pPD96.04. To generate tissue-specific NMD reporters and controls, each tissue-specific construct (25 ng µl$^{-1}$) and pRF4 (*rol-6(su1006)*) (75 ng µl$^{-1}$) were coinjected into Day 1 adult worms.

**Microscopy.** *smg-2::gfp; odr-1p::rfp* transgenic worms were mounted on a 2% agarose pad and anaesthetized by 100 mM sodium azide (DAEJUNG, Siheung, South Korea). Images of the transgenic worms were captured by using an AxioCam HRc CCD digital camera (Zeiss Corporation, Jena, Germany) connected to a Zeiss Axio Scope A1 compound microscope (Zeiss Corporation). Examination of DAF-16::GFP subcellular localization was performed as described previously with some modifications[37]. Worms were treated with indicated RNAi from eggs to L3 stage. The worms were then anaesthetized by using (−)-tetramisole hydrochloride (2 mM, Sigma) on a 2% agar pad before microphotograph images were captured.

**NMD GFP reporter analysis.** *lacZ(PTC)*, *sec-23p::gfp::lacZ(PTC)*, an NMD reporter[15], was used for analysing changes in NMD activity. *lacZ(WT)*, *sec-23p::gfp::lacZ* was used as a normalization control. Transgenic worms anaesthetized by using 100 mM sodium azide (DAEJUNG) were placed on a 2% agarose pad. Bright field and FITC filter images of the transgenic animals were captured by using an AxioCam HRc CCD digital camera (Zeiss Corporation, Jena, Germany) with a Zeiss Axio Scope A1 compound microscope (Zeiss). The head parts of the worms were used for quantification due to high levels of autofluorescence in the intestine of aged worms. The GFP intensity was normalized by subtracting the autofluorescence intensity values of age- and genotype-matched non-transgenic control animals. Quantitative analysis for the fluorescence levels was performed by using ImageJ (Java-based image processing programme, National Institutes of Health, http://rsbweb.nih.gov/ij). The represented images in Fig. 1b and Supplementary Fig. 1b were captured using the Nikon A1si/Ni-E upright confocal microscope with ×60, 1.4 NA oil-immersion objective lens. Confocal microscopy data were acquired in the Advanced Neural Imaging Center in Korea Brain Research Institute. Images of tissue-specific NMD reporters were obtained with the same method that was used for capturing the images of ubiquitous promoter-driven *lacZ(PTC)* animals with minor modifications. For RNAi-treated conditions, worms were treated with corresponding RNAi for one generation (Supplementary Fig. 1c,d) or two generations (Fig. 3a). The images of Day 1 adult worms were then captured. For tissue-specific NMD reporters, fluorescence levels of age-matched CF1290 that expressed only the *rol-6D* coinjection marker were subtracted to normalize an age-dependent increase in autofluorescence. For an intestine-specific NMD reporter, the anterior part of the intestine was quantified for the analysis.

**Dauer formation.** Dauer formation assay was performed as described previously with some modifications[17]. Briefly, several adult worms were allowed to lay eggs at 22.5 °C for 6 h and then removed. After 3 or 4 days, the number of dauers was counted using morphological criteria[38].

**Quantitative RT–PCR analysis.** Quantitative RT–PCR assay and analysis were performed as described previously[39]. All worms were grown at 20 °C except animals in *glp-1(e2141)* backgrounds, and their controls, which were grown at 25 °C until developing to Day 1 adults. Total RNA from synchronized Day 1 adult worms was extracted by using RNA Isoplus (Takara, Shiga, Japan). cDNA was generated by using a reverse transcription system (Promega, Madison, WI, USA), and was used for quantitative PCR for measuring the expression of each specific gene. The quantitative PCR with SYBR green dye (Applied Biosystems, Foster City, CA, USA) was performed by using StepOne Real Time PCR System (Applied Biosystems, Foster City, CA, USA) and analysed by using a comparative $C_T$ method. *ama-1* (ref. 39) or *pmp-3* (ref. 40) mRNA level was used as a control for normalization. See Supplementary Table 2 for the list of oligonucleotides used for the quantitative RT–PCR.

**RT–PCR analysis.** Bleached eggs of WT or *smg-2(qd101)* mutant animals were hatched and grown until reaching L4 stages. FUdR (bioWORLD, Ohio, USA) was then directly added on the NGM to a final concentration of 50 μM. Total RNAs were extracted from Days 1 and 9 of WT and *smg-2(qd101)* mutant worms. cDNAs were made by reverse transcription. *rpl-12* cDNA was amplified using the same primers that were used in a previous study[41]. PCR products were analysed using a 1.5% agarose gel. See Supplementary Table 3 for the list of oligonucleotides used for the RT–PCR.

**Body bending assays.** Body bending assays were performed as described previously[34]. Synchronized worms were transferred onto 5 μM FUdR (Sigma)-treated plates at Day 1 adult stage for experimental conditions for maintaining consistency with lifespan assays. The worms cultured on the plates were transferred into 1 ml M9 buffer-containing wells of 24-well plates. After 30 s of initial stabilization, body bending of the worms in each well was recorded by using DIMIS-M camera (Siwon Optical Technology, Anyang, South Korea). The body bends of individual worms were counted for 30 s and calculated for the number of body bends per minute.

**mRNA sequencing.** Worm samples for mRNA sequencing were prepared as described previously with some modifications[17]. Total RNA was extracted from Day 1 adults of WT, *daf-2(e1370)*, *smg-2(qd101)* and *smg-2(qd101); daf-2(e1370)* animals. The RNA samples were collected twice independently and used as duplicate samples for the following analysis. For all samples, RNA integrity numbers, which reflect the qualities of samples, were sufficiently high (≥9.9). For cDNA libraries, cDNAs were generated from mRNAs by reverse transcription. The cDNAs were sequenced using an Illumina HiSeq2500 150 bp paired end platform (Macrogen Inc., South Korea).

**mRNA sequencing analysis.** Raw sequencing reads were examined for quality using FASTQC. Sequences relevant to 3′ adapters were trimmed with cutadapt[42]. The resulting trimmed reads were aligned to the *C. elegans* genome build ce10 (WS220) using TopHat 2.0.14 (ref. 43) and bowtie2-2.2.5 (ref. 44) with default parameters. The reads mapped to potential exons as well as splice junctions were assembled into transcripts. The transcripts were annotated by using ENSEMBL (release 65). The expression levels of annotated transcripts were measured by fragments per kilobase per million mapped reads[45] using Cufflinks 2.2.1 (ref. 46). Differentially expressed genes between conditions were tested by using cuffdiff. RNA types were annotated with UCSC genome browser (WS220/ce10). Transcripts annotated as protein-coding RNAs were then included in our further analysis, whereas other RNA species (rRNAs, tRNAs, snRNAs, snoRNAs, ncRNAs and pseudogene RNAs) were excluded. Transcripts whose status had at least one term of NOTEST (not enough alignments for testing), HIDATA (too many fragments in locus) or FAIL, and transcripts whose fragments per kilobase per million mapped reads were zero were excluded from further analysis (http://cole-trapnell-lab.github.io/cufflinks/). Neuronal IIS[24] and *smg-1* (ref. 13) transcriptome data were analysed with the same method that was used for analysing our sequencing data. For neuronal IIS transcriptome data analysis[24], we removed low-quality samples.

**Analysis of transcripts with PTCs.** PTCs were defined using the Bioconductor package SpliceR 1.14.0 with default parameters[22]. Briefly, PTCs were annotated when stop codons were located more than 50 nt upstream of the final exon–exon junctions.

**Analysis of uORFs.** Potential ORFs were estimated from the assembled transcripts using TransDecoder 2.0.1. Only 'complete' ORFs with intact start codons and stop codons were chosen for further analysis. The locations and sequences of predicted

5′ UTRs were then extracted. The presence of ORFs in the 5′ UTRs was predicted using 'getorf' command in EMBOSS 6.6.0 with -minsize 3 -find 3. The longest ORFs in the frame of CDS were identified as 'uORFs'.

**Analysis of 3′ UTR lengths.** The lengths of 3′ UTRs were calculated from the ends of ORFs estimated by TransDecoder 2.0.1 and the ends of the assembled transcripts. Alternative 3′ UTR annotation was conducted by using IsoSCM 2.0.11 (ref. 47). Combining results of these two methods enabled more accurate estimation of 3′ UTR lengths. Plots were generated using R version 3.1.3 (www.r-project.org) and Bioconductor version 3.1 (www.bioconductor.org).

**mRNA half-life assays.** mRNA half-life assays using L1 stage worms were performed as described previously with some modifications[48]. Eggs were obtained by bleaching gravid adults, and the eggs were incubated in M9 buffer solution overnight. Synchronized L1 worms ($10^4$ worms per ml) were cultured with *Escherichia coli* OP50 for 2 h in S medium[49] with constant shacking. A transcription inhibitor 5,6-Dichloro-1-beta-D-ribofuranosylbenzimidazole (DRB, Sigma) was dissolved in dimethyl sulfoxide (Junsei, Tokyo, Japan). After the 2 h of culture, worms were treated with DRB at various final concentrations (100, 200, 300, 500 and 1,000 μg ml$^{-1}$) with 1% of dimethyl sulfoxide for optimization. We noted that precipitation was observed at the maximum concentration of DRB (1,000 μg ml$^{-1}$). The worms were then collected at each time point, and washed with M9 buffer three times. Total RNA from the DRB-treated worms was extracted by using RNA Isoplus (Takara), and the half-life of mRNA was measured by qRT–PCR. 18S rRNA level was used as a normalization control[48]. For mRNA half-life assays using adult worms, synchronized worms were cultured on solid media until they reached Day 1 adults. The Day 1 adults were washed and transferred to S medium, and then incubated for 1 h with OP50 with constant shacking. Subsequently, DRB was added to a desired concentration (300 or 500 μg ml$^{-1}$). The DRB-treated adult worms were collected at each time point, and washed with M9 buffer three times.

**Statement for data analyses.** The authors declare that sample size was chosen based on literature studies in this field without statistical method, and no randomization or blind test was used for sample analyses.

**Data availability.** RNA-Seq data have been deposited in Gene Expression Omnibus under accession code GSE94077.

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

## Acknowledgements

Drs Cynthia Kenyon, Gary Ruvkun, Dennis Kim and Javier F. Cáceres, and the Caenorhabditis Genetics Center provided some of the *C. elegans* strains used in this study. We thank all Lee and Nam laboratory members for help and discussion. This work was supported by a grant of the Korean Health Technology R&D Project, Ministry of Health and Welfare (HI14C2337) and by POSCO Green Science Project to S.-J.V.L., by Institute for Basic Science (IBS-R013-D1) to H.G.N. and by Korea Brain Research Institute (KBRI) basic research programme through KBRI funded by the Ministry of Science, ICT & Future Planning (17-BR-01) to C.M.H. Additional data are available in the supplementary materials.

## Author contributions

H.G.S., M.S., H.G.N. and S.-J.V.L. designed the experiments. H.G.S., M.S., W.H., D.L., S.W.A.A., M.A., K.S., R.N.A., Y.R., C.M.H. and S.-J.V.L. performed the experiments. H.G.S., M.S., Y.K.K., R.K., S.H., T.-Y.R. and S.-J.V.L. analysed the data. H.G.S., C.T.M., H.G.N. and S.-J.V.L. wrote the manuscript.

## Additional information

**Competing financial interests:** The authors declare no competing financial interests.

