## [Peer Review File · Nature Communications]

Reviewer #1 (Remarks to the Author)

In this manuscript, Son et al address an important and intriguing question: does maintenance of RNA quality determine aging. The authors made several important discoveries. First, they show that mutations in the IIS extend lifespan through modulation of NMD activity. Second, they showed that NMD acts mainly in the nervous system of the animals to extend lifespan. These findings could have a big impact on several fields such as aging, neurodegeneration, etc. However, the papers should address several points that will strengthen their conclusions.

Major points:

- In Figure 1, the reporter experiment is not sufficient to conclude that NMD activity decreases with age. Other experiments can strengthen this conclusion (accumulation of endogenous NMD targets, etc). Furthermore, mutations in *smg-2* do not shorten lifespan in the wild-type background. Thus, NMD does not seem to be important for normal aging.
- *smg-2* is required for lifespan extension of germline-lacking worms and IIS mutants. Since *daf-16* is required for the longevity phenotype of both *daf-2* and *glp-1* mutants, the effects of *smg-2* could depend on *daf-16* activation. Does *daf-16* regulate the activity or expression of *smg-2*?
- I am not convinced that the data showed in figures 4a-d completely supports the authors' conclusions regarding the importance of *smg-2* in IIS mutants. It seems that mutations in *smg-2* in the *daf-2* background does not change the percentage of uORF+ and PTC+ transcripts in the up-regulated transcripts (as observed in the wild-type background in Supplementary Figure 5). However, it's clear the effect in the percentage of uORF/PTC+ transcripts in the down-regulated transcripts. Can the authors discuss this in the main text? Another possibility would be to test whether knockdown of *smg-1* or other NMD-related genes increase the percentage of uORF+ and PTC+ transcripts in the up-regulated transcripts of the *daf-2* mutants.
- Does the effects observed in figure 4g-h depend on *smg-2* activity?

Minor points:

- The authors should mention in the main text that they examined several endogenous NMD targets by qRT-PCR.
- The authors should give more details about the reporter of NMD activity in the main text for readers not familiarized with this system.
- The authors mention in the abstract that NMD activity promotes healthy aging. However, they only focus on lifespan experiments. Thus, they should change the abstract or perform healthspan experiments.

Reviewer #2 (Remarks to the Author)

The authors sought to demonstrate that aging decreases NMD activity and insulin signaling is required to regulate NMD activity. The authors conclude that maintained NMD activity is important mainly in the nerve system for the lifespan extension in *daf-2* and *glp-1* mutants but not that of *eat-2*, *vhl-1* and *isp-1* mutants. The authors also conclude that loss of NMD activity changes levels of mRNAs that are up or down-regulated in *daf-2* mutants. The authors concluded that the insulin pathway regulates RNA surveillance through NMD to control aging.

Although the data are interesting, the manuscript has significant weaknesses that prevent strong conclusion. Major revisions would be necessary before publication.

1. The basic finding that NMD declines with age is not completely convincing. In Figure 1, the increase in fluorescence is small and it is not clear from the methods section how increased autofluorescence in aged animals was controlled for. In addition, the reporter transgene should be tested under tissue-specific promoters. This would also allow determination of where age is

affecting NMD and in particular establish whether this is occurring in neurons. Finally, it should be possible to look at splicing of endogenous NMD targets like rpl-12 in aged animals, or to do a global analysis.

2. The conclusion that NMD is specifically required for increased longevity in animals with defective insulin signaling or no germline, but not in other models, is done using smg-2 RNAi. Yet comparing fig 2a and 2b, smg-2 RNAi has only a minor effect compared to smg-2 mutants. Further, given that the paper argues the primary locus of NMD activity is in neurons, and that neurons are often resistant to RNAi, these data are very difficult to interpret.

In addition, the NMD sensor should be examined in the insulin and germline backgrounds-and for insulin, the effect of daf-16 on the daf-2 phenotype should be tested.

3. The finding that different smg genes have different effects on increased longevity in daf-2 is very difficult to understand. Do these genes have different effects on NMD? These data raise the possibility that the longevity effects of the smg genes is due to some other function than NMD.

4. Figure 3 a-d is very confusing. What is being compared to drive the conclusion that daf-2 reduces the expression of uORF genes in a smg-dependent manner? Why is there a smg dependence in the down genes, but not in the up genes?

5. To complement the RNAi experiments, tissue-specific rescue of smg-2 should be done. It would also be nice to determine whether overexpression of smg-2 can increase lifespan.

Reviewer #3 (Remarks to the Author)

The authors conclude that while SMG2 deficiency has little effect on WT worms - or a dietary restriction mimetic, a hif-1 gain-of-function mutant or a mitochondrial respiration-defective mutant - it substantially shortens the long lifespan induced by genetic inhibition of daf2 insulin/IGF-1 receptor. This is true for neurons but not true for intestine-, muscle- or hypodermis-specific SMG2 RNAi, leading the authors to conclude that the maintenance of NMD in neurons is critical for long lifespans of worms with reduced IIS.

While interesting and certainly with important potential, there are problems with this submission, as noted in the itemized review below. Primarily, the authors have not convincingly demonstrated that IIS enhances NMD; if it does, then it seems important to show something about mechanism - how is surveillance enhanced?

Specific Comments

Abstract, line 8. Here, the authors fail to mention an important aspect of NMD - in the decay of an estimated 10% of normal physiologic transcripts. They also do not discuss how NMD is regulated conditionally by cells to adapt to changing environments, e.g. possibly the one created in response to changes in IGF-1 receptor activity and/or as cells age.

Abstract Line 9-11: "We found that long-lived daf-2...mutants displayed enhanced NMD activity...". This is a central argument of the paper, but hasn't been shown convincingly. As far as I can tell, only a "fingerprint" of downregulated (PTC+, uORF-containing but unexpectedly not "long 3'UTR") transcripts found in daf-2 mutant worms is used to make this conclusion. The reporter from Figure 1 could be used in a daf-2 background (combined with appropriate half-life studies that show reduction in half-life in the daf-2 background) to make the argument more convincing.

Page 3, line 7. "Ribostasis" is not a word known by this reviewer to be generally used by RNA biologists and is confusing since, a priori, it could be interpreted to mean ribosome homeostasis -

"RNAstasis" would be much better, but again is not a term that is used. This reviewer recommends simply stating "RNA homeostasis", which everyone will understand.

Figure 1 and page 3, line 16 of conclusion. How much of the increase in mRNA abundance when SMG2 is downregulated is transcriptional vs. due to an increase in mRNA half-life? Until this is known, the authors cannot comment on the efficiency of NMD. Also related: The natural decline in NMD activity with age would seem to be a central argument for relevance of the story. The only real evidence for this is Figure 1d, but the data are confounded by the possibility that the activity of the Sec23p promoter used to drive the reporter changes with age. Half-life studies can resolve this.

Figures 1 and 2. Since *C. elegans* SMG2, which is orthologous to human/mammalian UPF1, is used in many pathways other than NMD, at least in human/mammalian cells, the authors would do well to vary the abundance of another NMD factor. In fact, the authors later show that shorter life-spans typify worms that lack SMG-2 relative to worms lacking one of several other NMD factors (Figure 2g). How does the efficiency of NMD compare in worms deficient in the various NMD factors? This is a key issue that needs to be resolved before the authors can conclude what they do.

Page 5, line 3. Why do *daf-2* mutations reduce the level of mRNAs with an uORF or PTC but not "long" (what do the authors mean by "long"?) 3'UTR? The suggestion that *daf-2* mutations promote NMD is not particularly solid - can the authors reproduce this finding using various types of NMD reporters?

Page 6. Does IIS, i.e. insulin/IGF (insulin-like growth factor)-like signaling, really regulate RNA surveillance through NMD? This reviewer is not convinced.

Minor Comment

Abstract, line 7. RNA is the second component in the flow of information (Central Dogma):
DNA → mRNA → protein

Reviewers' comments:

Reviewer #1 (Expert in worm ageing; Remarks to the Author):

In this manuscript, Son et al address an important and intriguing question: does maintenance of RNA quality determine aging. The authors made several important discoveries. First, they show that mutations in the IIS extend lifespan through modulation of NMD activity. Second, they showed that NMD acts mainly in the nervous system of the animals to extend lifespan. These findings could have a big impact on several fields such as aging, neurodegeneration, etc. However, the papers should address several points that will strengthen their conclusions.

Major points:

- In Figure 1, the reporter experiment is not sufficient to conclude that NMD activity decreases with age. Other experiments can strengthen this conclusion (accumulation of endogenous NMD targets, etc).

We appreciate this reviewer's comment. To strengthen our conclusion regarding the age-dependent decrease in NMD activity, we performed RT-PCR to detect endogenous PTC-containing (*rpl-12(PTC)*) and PTC-negative *rpl-12* (*rpl-12(WT)*) transcripts simultaneously using day 1 (young) and day 9 (old) worm samples. We found that the ratio of *rpl-12(PTC)/rpl-12(WT)* transcripts was slightly but significantly increased during aging. We describe this finding in Results as follows,

Results, page 4, line 64: "Next, we examined whether the mRNA levels of PTC-containing *rpl-12* (*rpl-12(PTC)*), a well-established endogenous target of NMD (Supplementary Fig. 1e) (Mitrovich and Anderson, 2000), changed during aging. We found that old animals displayed increased *rpl-12(PTC)* levels when they were normalized with *rpl-12(WT)* transcripts that do not contain PTC, compared to that in young worms (Fig. 1e). This result supports the idea that NMD activity decreases during aging"

In addition, we performed experiments using four different tissue-specific NMD reporters. We found that GFP intensities of hypodermis-, muscle- and intestine-specific promoter-driven NMD reporters increased during aging, while those of neuron-specific promoter-driven NMD reporter did not (Fig 1f). This indicates that NMD activity in neurons was maintained longer than in other tissues. The changes in the GFP levels appear to be independent of promoters, because we obtained similar results with NMD reporters using total four different promoters. We revised the Results and Methods of our manuscript as follows,

Results, page 4, line 70: “We sought to determine in which tissues NMD activity changed during aging by examining neuron-, hypodermis-, muscle-, and intestine-specific NMD reporters. We found that the normalized GFP intensities of *lacZ(PTC)/lacZ(WT)* were increased during aging in the hypodermis, muscle, and intestine, while those in neurons were largely unaffected with age (Fig 1f). This result raises a possibility that the maintenance of NMD activity in neurons is sustained longer than that in other tissues.”

Furthermore, mutations in *smg-2* do not shorten lifespan in the wild-type background. Thus, NMD does not seem to be important for normal aging.

We thank the reviewer for raising this point. We agree with the reviewer’s comment that NMD may not be a limiting factor for normal aging. We made this point in our text as follows,

Results, page 4, line 77: “Loss of *smg-2* had a small effect on wild-type lifespan, suggesting that NMD may not limit normal lifespan. However, *smg-2* mutation or RNAi substantially shortened the long lifespan induced by genetic inhibition of the *daf-2/insulin/insulin-like growth factor 1 (IGF-1) receptor* (Fig. 2a-c, Supplementary Fig. 2a,b).”

Interestingly, we obtained new data showing that overexpression of SMG-1, which activates SMG-2 via phosphorylation, increased lifespan. These results suggest that upregulation of NMD can delay normal aging. We now added these new data in the text,

Results, page 7, line 148: “...overexpression of SMG-1, a protein kinase that phosphorylates and activates SMG-2, increased lifespan (Fig. 5e).”

- *smg-2* is required for lifespan extension of germline-lacking worms and IIS mutants. Since *daf-16* is required for the longevity phenotype of both *daf-2* and *glp-1* mutants, the effects of *smg-2* could depend on *daf-16* activation. Does *daf-16* regulate the activity or expression of *smg-2*?

> We appreciate the reviewer's critical comments. To examine whether *smg-2* expression levels were changed by *daf-2* or *glp-1* mutations in a *daf-16*-dependent manner, we performed qRT-PCR analysis. We found that *smg-2* mRNA levels were not changed by *daf-16(mu86)*, *daf-2(e1370)*, *daf-16(mu86); daf-2(e1370)*, *glp-1(e2141)*, or *daf-16(mu86); glp-1(e2141)* mutations (Supplementary Fig. 5a,b). We also found that the GFP intensity of *smg-2p::smg-2::gfp* was not changed by *daf-16* RNAi treatment (Supplementary Fig. 5c). These results suggest that the expression of *smg-2* is independent of *daf-2*, *glp-1* or *daf-16*.

As mentioned above, we obtained new data showing that overexpression of SMG-1, which activates SMG-2 via phosphorylation, increased lifespan. This result implies that post-transcriptional regulation of SMG-2 through phosphorylation by SMG-1 may exert its lifespan-extending effects. We now added these new data in the text,

Results, page 5, line 93: “A FOXO homolog, DAF-16 is a well-known longevity transcription factor acting downstream of two of the mutants, *daf-2(-)* and *glp-1(-)*, which we showed to require *smg-2* for their longevity (Hsin and Kenyon, 1999; Kenyon, 2010). We therefore examined the possibility that *smg-2* expression was changed by *daf-2* or *glp-1* mutations in a *daf-16*-dependent manner. We found that the mRNA level of *smg-2* was not changed by *daf-2*, *glp-1*, or *daf-16* mutations (Supplementary Fig. 5a,b), or the GFP-fused SMG-2 level was not changed by *daf-16* RNAi (Supplementary Fig. 5c).”

Results, page 7, line 147: “*smg-2* overexpression did not affect the lifespan of wild-type animals (Fig. 5d). In contrast, overexpression of *smg-1*, which encodes a protein kinase that activates SMG-2 via phosphorylation, increased lifespan (Fig. 5e). With the finding that *smg-2* mRNA level was not changed by *daf-2*, *glp-1* or *daf-16* mutations (Supplementary Fig. 5b,c), these data indicate that post-transcriptional regulation of SMG-2 via phosphorylation by SMG-1 may exert its lifespan-extending effects.”

In addition, we performed converse experiments by asking whether knockdown of *smg-2*

gene affected DAF-16 activity by observing DAF-16 nuclear localization and dauer formation. Unexpectedly, we found that *smg-2* RNAi increased the nuclear localization of DAF-16::GFP in wild-type and *daf-2(e1368)* backgrounds (Supplementary Fig. 6a,b). *smg-2* mutations also further increased dauer formation of *daf-2(e1370)* mutants (Supplementary Fig. 6c). Thus, genetic inhibition of *smg-2* appears to increase DAF-16 activity, possibly as a compensatory mechanism, when the longevity response of *daf-2* mutants was decreased by the inhibition of NMD. We described these results in the text as follows,

Results, page 5, line 98: “Conversely, we examined whether DAF-16 activity was changed upon inhibition of *smg-2*. Unexpectedly, we found that *smg-2* RNAi increased the nuclear localization of DAF-16::GFP in wild-type and *daf-2(e1368)*, a weak *daf-2* mutant allele amenable for the detection of an increase in nuclear DAF-16 (Supplementary Fig. 6a,b). *smg-2* mutations also further increased dauer formation, which requires up-regulation of DAF-16 (Hu, 2007), in *daf-2(e1370)* mutants (Supplementary Fig. 6c). Thus, the genetic inhibition of *smg-2* appears to increase DAF-16 activity, possibly as a compensatory mechanism when the longevity response of *daf-2* mutants was decreased by the inhibition of NMD.”

- I am not convinced that the data showed in figures 4a-d completely supports the authors' conclusions regarding the importance of *smg-2* in IIS mutants. It seems that mutations in *smg-2* in the *daf-2* background does not change the percentage of uORF+ and PTC+ transcripts in the up-regulated transcripts (as observed in the wild-type background in Supplementary Figure 5). However, it's clear the effect in the percentage of uORF/PTC+ transcripts in the down-regulated transcripts. Can the authors discuss this in the main text? Another possibility would be to test whether knockdown of *smg-1* or other NMD -related genes increase the percentage of uORF+ and PTC+ transcripts in the up-regulated transcripts of the *daf-2* mutants.

> We thank the reviewer for raising this issue. First, we think this reviewer's question regarding 4a-d was actually about figure 3a-d of the previous version of the manuscript. If that is the case, the reason for the percentage of PTC- and uORF-containing transcripts was increased among the down-regulated transcripts is because the transcriptome of *daf-2* mutants

was compared to that of *smg-2(-); daf-2(-)* mutants. If we compare the changes in *smg-2(-); daf-2(-)* to *daf-2(-)*, the percentage of PTC- and uORF-containing transcripts was increased in up-regulated transcripts as we observed in *smg-2(-)* to WT. We admit that the previous comparison between *daf-2(-)* and *smg-2(-); daf-2(-)* was somewhat confusing. Therefore, we changed the order of comparison from “*daf-2(-)* with *smg-2(-); daf-2(-)*” to “*smg-2(-); daf-2(-)* with *daf-2(-)*”. In addition, we now think it is more appropriate that we should analyze the *smg-2* dependency of the transcripts that were down-regulated in *daf-2(-)* mutants compared to WT, instead of comparing all the transcripts in *daf-2(-)* with those in *smg-2(-); daf-2(-)*. Through this new analysis, we obtained consistent results with our original conclusion that levels of PTC- or uORF-containing targets were increased by *smg-2* mutations (Fig 4d,e). In addition, when we analyzed the down-regulated transcripts in *daf-2* mutants compared to wild-type, *smg-2* mutations increased the expressions of long 3’UTR-containing transcripts as well (Fig. 4f). This changed our original conclusion but actually further strengthened our point that *daf-2* mutations decrease the levels of various NMD target transcripts. Using these new analyses, we modified the Results as follows,

Results, page 7, line 131: “Next we examined whether the expression of PTC-, uORF-, or long 3’UTR-containing transcripts that were down-regulated in *daf-2* mutants compared to wild-type animals were changed in *smg-2; daf-2* mutants. The expression levels of down-regulated PTC-, uORF-, or 3’UTR-containing transcripts in *daf-2* mutants were increased in *smg-2(-); daf-2(-)* mutants (Fig. 4d-f).”

- Does the effects observed in figure 4g-h depend on *smg-2* activity?

> This is an important point but we believe it is beyond the scope of the current paper, which focused on the importance of RNA surveillance for longevity. We want to address this as a separate paper in a future study by analyzing neuron-specific transcriptomes of *smg-2; daf-2* mutants.

Minor points:

- The authors should mentioned in the main text that the examined several endogenous NMD targets by qRT-PCR.

> We thank the reviewer for making this point and describe the results as follows,

Results, page 7, line 136: “We found that the mRNA level of PTC-containing *rpl-12* measured by using qRT-PCR was decreased in *daf-2* mutants, in a *smg-2*-dependent manner (Fig. 4g,h).”

- The authors should give more details about the reporter of NMD activity in the main text for readers not familiarized with this system.

> We thank the reviewer for this thoughtful comment. We describe the details of the NMD reporter in the main text as follows,

Results, page 4, line 57: “This NMD reporter contains a premature termination codon (PTC) in the first exon of *lacZ* fused with GFP, *sec-23p::gfp::lacZ(PTC)*. Therefore, in normal conditions, this transgene is degraded by NMD pathway, resulting in dim GFP. In contrast, when NMD activity is decreased or blocked, GFP intensity is increased.”

- The authors mention in the abstract that NMD activity promotes healthy aging. However, they only focus on lifespan experiments. Thus, they should change the abstract or preform healthspan experiments.

> We thank the reviewer for raising this issue. We changed “healthy aging” to “longevity” in the Abstract as follows,

Abstract, page 2, line 22: “Here, we show that nonsense-mediated mRNA decay (NMD), which degrades abnormal transcripts as well as some normal transcripts, promotes longevity by enhancing RNA surveillance in *C. elegans*.”

Although we downplayed healthspan in the Abstract, we still determined age-dependent changes in the motility (body bends in liquid), a popular assay used for measuring healthspan, with wild-type, *smg-2(qd101)*, *daf-2(e1370)*, and *smg-2(qd101); daf-2(e1370)*. We found that delayed age-dependent declines in the motility conferred by *daf-2(e1370)* mutations were

partially suppressed by *smg-2(qd101)* mutations. We added the data and described them as follows,

Figure 2 legend, page 26, line 631: “*smg-2* mutations also partially suppressed delayed age-dependent declines in body movement of *daf-2* mutants (Supplementary Fig. 3).”

Reviewer #2 (Expert in worm ageing; Remarks to the Author):

The authors sought to demonstrate that aging decreases NMD activity and insulin signaling is required to regulate NMD activity. The authors conclude that maintained NMD activity is important mainly in the nerve system for the lifespan extension in *daf-2* and *glp-1* mutants but not that of *eat-2*, *vhl-1* and *isp-1* mutants. The authors also conclude that loss of NMD activity changes levels of mRNAs that are up or down-regulated in *daf-2* mutants. The authors concluded that the insulin pathway regulates RNA surveillance through NMD to control aging.

Although the data are interesting, the manuscript has significant weaknesses that prevent strong conclusion. Major revisions would be necessary before publication.

1. The basic finding that NMD declines with age is not completely convincing. In Figure 1, the increase in fluorescence is small and it is not clear from the methods section how increased autofluorescence in aged animals was controlled for.

> We thank the reviewer for this valuable comment. We previously took pictures of the head parts of the worms to reduce the effects of autofluorescence. As we agree with the reviewer that it was not a sufficient control for autofluorescence, we obtained the microphotographs of NMD reporters again using proper age- and genotype-matched controls. For example, when we obtained the pictures of NMD reporter animals with different ages, we also obtained the pictures of age- and genotype-matched non-transgenic control animals with the same exposure. We then normalized the changes in the GFP intensity of NMD reporter animals

with that of the age- and genotype-matched non-transgenic controls, which reflect autofluorescence. This new analysis did not change our original conclusion. We added this information in the Methods.

Methods, page 15, line 318: “The GFP intensity was normalized by subtracting the autofluorescence intensity values of age-and genotype-matched non-transgenic control animals.”

Methods, page 15, line 329: “For tissue-specific NMD reporters, fluorescence levels of age-matched CF1290 that expressed only the *rol-6D* coinjection marker were subtracted to normalize an age-dependent increase in autofluorescence.”

In addition, the reporter transgene should be tested under tissue-specific promoters. This would also allow determination of where age is affecting NMD and in particular establish whether this is occurring in neurons.

> We appreciate this excellent suggestion by the reviewer. We generated *dpy-7p*, *ges-1p*, *myo-3p*, and *rgef-1p*-driven NMD reporter transgenic worms for determining hypodermal, intestinal, muscle-specific, and neuronal roles of NMD, respectively. We also generated four control tissue-specific transgenic worms expressing GFP-lacZ fusion transgenes without premature termination codons. We found that the hypodermal, muscle-specific and the intestinal GFP levels increased during aging, while that of neuronal NMD reporter did not (Fig 1f). We added our new data and revised Results.

Results, page 4, line 70: “We sought to determine in which tissues NMD activity changed during aging by examining neuron-, hypodermis-, muscle-, and intestine-specific NMD reporters. We found that the normalized GFP intensities of *lacZ(PTC)/lacZ(WT)* levels were increased during aging in the hypodermis, muscle, and intestine, while those in neurons were largely unaffected with age (Fig 1f). This result raises a possibility that the maintenance of NMD activity in neurons is sustained longer than that in the other tissues.”

Finally, it should be possible to look at splicing of endogenous NMD targets like *rpl-12* in

aged animals, or to do a global analysis.

> We thank the reviewer for this comment. To examine PTC-containing *rpl-12* transcript level we used RT-PCR for quantifying PTC-containing (*rpl-12(PTC)*) and PTC-negative transcripts (*rpl-12(WT)*) at the same time with the same primers. We detected a small but significant increase in *rpl-12(PTC)/rpl-12(WT)* transcript ratio in aged worms (day 9) compared to that in young worms (day 1) (Fig. 1e). Thus, we revised our manuscript as follows.

Results, page 4, line 64: “Next, we examined whether the mRNA levels of PTC-containing *rpl-12* (*rpl-12(PTC)*), a well-established endogenous target of NMD (Supplementary Fig. 1e) (Mitrovich and Anderson, 2000), changed during aging. We found that old animals displayed increased *rpl-12(PTC)* levels when they were normalized with *rpl-12(WT)* transcripts that do not contain PTC, compared to that in young worms (Fig. 1e). This result supports the idea that NMD activity decreases during aging.”

2. The conclusion that NMD is specifically required for increased longevity in animals with defective insulin signaling or no germline, but not in other models, is done using *smg-2* RNAi. Yet comparing fig 2a and 2b, *smg-2* RNAi has only a minor effect compared to *smg-2* mutants. Further, given that the paper argues the primary locus of NMD activity is in neurons, and that neurons are often resistant to RNAi, these data are very difficult to interpret.

> We thank and agree with the reviewer's very important comment. Because of the RNAi efficiency problem, we generated and measured the lifespan of following double mutants: *smg-2(qd101); isp-1(qm150)*, *smg-2(qd101); eat-2(ad1116)*, *smg-2(qd101); vhl-1(ok161)*, and *smg-2(qd101); osm-5(p813)*. We found that *smg-2* mutations suppressed the longevity of *isp-1* (Fig. 2d) and *eat-2* (Fig. 2e) mutants but not that of *vhl-1* (Fig. 2g) or *osm-5* (Fig. 2h) mutants, which is different from our previous RNAi results. Excitingly, these new data suggest that NMD is an important regulator of several longevity mechanisms. We have changed the Title, Abstract, and the main text to include this point. We also appreciate the reviewer's excellent point regarding the RNAi resistance of *C. elegans* neurons, as our new results are consistent with the idea that neuronal *smg-2* is crucial for longevity. We now

revised our text to describe these new data as follows,

Title: From “Reduced insulin/IGF-1 signaling promotes longevity via enhancing RNA surveillance in *C. elegans*”

=> To “RNA surveillance via nonsense-mediated mRNA decay is crucial for longevity in *C. elegans*”

Abstract, page 2, line 26: “NMD components, including *smg-2/UPF1*, were required for the long lifespan of *daf-2* mutants. In addition, *smg-2/UPF1* was required for dietary restriction (*eat-2*), germline loss (*glp-1*), and mitochondrial mutant (*isp-1*) longevity, suggesting that NMD is generally important for maintenance of biological function with age.”

Results, page 4, line 76: “As we found that NMD activity generally declined during aging, we wondered whether NMD affects organismal lifespan. Loss of *smg-2* had a small effect on wild-type lifespan, suggesting that NMD may not limit normal lifespan. However, *smg-2* mutation or RNAi substantially shortened the long lifespan induced by genetic inhibition of the *daf-2*/insulin/insulin-like growth factor 1 (IGF-1) receptor (Fig. 2a-c, Supplementary Fig. 2a,b). We found that *smg-2* mutations also shortened the longevity of mitochondrial respiration-defective *isp-1*(-) mutants (Fig. 2d), dietary restriction mimetic *eat-2*(-) mutants (Fig. 2e), and germline-deficient *glp-1*(-) mutants (Fig. 2f). In contrast, *smg-2* mutations had no effect on the long lifespan of the *hif-1* gain-of-function *vhl-1*(-) mutants (Fig. 2g), or sensory-defective *osm-5*(-) mutants (Fig. 2h). These data indicate that *smg-2* is required for lifespan extension rather generally but not universally. In contrast to the results with genetic mutants, *smg-2* RNAi did not decrease longevity induced by *isp-1* (Supplementary Fig. 2c), *eat-2* (Supplementary Fig. 2d), or *glp-1* (Supplementary Fig. 2e) mutations. The specific longevity-suppressing effect of *smg-2* RNAi on *daf-2* mutants is likely because *daf-2* mutants are RNAi sensitive (Kaletsky et al., 2015; Wang and Ruvkun, 2004). In addition, the stronger effects of *smg-2* mutations on longevity than those of *smg-2* RNAi raise a possibility that neuronal *smg-2* may play a major lifespan-regulatory role in longevity mutants, as neurons are often refractory to RNAi effects (Kamath et al., 2001; Timmons et al., 2001).”

In addition, the NMD sensor should be examined in the insulin and germline backgrounds-and for insulin, the effect of *daf-16* on the *daf-2* phenotype should be tested.

> This is a very critical point. We have been generating NMD reporters in various mutant backgrounds. We were able to generate and characterized the NMD reporter in a *daf-2* mutant background, the most important one, during the revision period. We crossed the *daf-2(e1370)* worms with the NMD reporter, *sec-23p::gfp::lacZ(PTC)(lacZ(PTC))*, and also generated control worms that do not harbor PTC, *sec-23p::gfp::lacZ(lacZ(WT))*. When we normalized the NMD reporter's GFP intensity with control, we found that both young and old *daf-2* mutants displayed a decrease in the GFP intensity of *lacZ(PTC)/lacZ(WT)* compared to wild-type animals (Fig. 4i). These data suggest that *daf-2* mutations up-regulates NMD.

Results, page 7, line 138: "In addition, *daf-2* mutations decreased the fluorescence intensity ratio of *lacZ(PTC)/lacZ(WT)* GFP reporter, both in young and old animals (Fig. 4i). Together, these data raise the possibility that decreased levels of PTC-, uORF-, or long-3'UTR-containing transcripts in *daf-2* mutants may contribute to longevity."

3. The finding that different *smg* genes have different effects on increased longevity in *daf-2* is very difficult to understand. Do these genes have different effects on NMD? These data raise the possibility that the longevity effects of the *smg* genes is due to some other function than NMD.

> We very much appreciate the reviewer's comment. We examined the GFP levels of the NMD reporter upon knocking down *smg-1* through *smg-7*. We found that knockdown of *smg-1* through *smg-5* significantly increased the GFP intensity, suggesting that NMD activity was indeed decreased by knocking down each of these NMD components (Fig. 3a). In contrast, *smg-6* or *smg-7* RNAi treatment had no effect on the GFP intensity (Fig. 3a). We also noticed that the GFP intensity extent was variable among *smg-1* through *smg-5* RNAi conditions. These different effects of *smg* genes on their target degradation are consistent with a previous report (Johns et al., 2007). Because of these results, we focused our lifespan analysis description on *smg-1* through *smg-5*. We describe these results as follows,

Results, page 6, line 107: “The NMD pathway is regulated by the NMD complex, composed of many components (SMG-1 through SMG-7). We found that knockdown of *smg-1* through *smg-5* significantly increased the GFP intensity of the NMD reporter, but *smg-6* and *smg-7* RNAi did not (Fig. 3a). We also noticed that the effects of *smg-1* through *smg-5* RNAi on the GFP intensity were variable (Fig. 3a). These data are consistent with reports showing that NMD components play both overlapping and distinct roles, and their effects on a same target can be variable (Johns et al., 2007). We also found that RNAi knockdown of each of *smg-1*, -2, -3, and -5, significantly suppressed the long lifespan of *daf-2* mutants (Fig. 3b-d,f); *smg-4* RNAi decreased the long lifespan of *daf-2* mutants in one out of two trials (Fig. 3e). Thus, NMD appears to promote longevity in animals with reduced insulin/IGF-1 signaling.”

Although we obtained negative data using *smg-6* RNAi and *smg-7* RNAi for the NMD reporter, we still included our lifespan results using *smg-6* and *smg-7* RNAi in the Supplementary Figure 4 and discussed the negative data in Figure legends.

Supplementary Fig. 4a legend, page 8, line 2: “*smg-6* RNAi shortened the lifespan of *daf-2* mutants in 1 out of 2 trials.”

Supplementary Fig. 4b legend, page 8, line 3: “*smg-7* RNAi did not affect the lifespan of wild-type or *daf-2* mutant animals. Because *smg-7* RNAi did not affect the GFP intensity of NMD reporter (Fig. 3a) or lifespan, we cannot exclude the possibility that *smg-7* RNAi treatment did not effectively knock down *smg-7* gene.”

4. Figure 3 a-d is very confusing. What is being compared to drive the conclusion that *daf-2* reduces the expression of uORF genes in a smg-dependent manner?

> We agree with the reviewer that our explanation was not sufficient. Previously we compared *smg-2* dependency using all the transcripts in *daf-2* and *smg-2*; *daf-2* mutants. However, it was not suitable for showing that the NMD target transcripts that were down-regulated by *daf-2* mutations were increased by *smg-2* mutations in *smg-2*; *daf-2* double mutants. Thus, we changed our analysis to determine the expression levels of NMD targets such as PTC- or uORF-containing transcripts, focusing on the genes whose mRNA levels

were decreased by *daf-2* mutations. We found that the expression levels of down-regulated NMD targets by *daf-2* mutations were increased in *smg-2; daf-2* mutants compared to *daf-2* mutants (Fig. 4d,e). Furthermore, when we analyzed the down-regulated transcripts in *daf-2* mutants compared to wild-type, *smg-2* mutations increased the expression of long 3'UTR-containing transcripts as well (Fig. 4f). This new analysis did not change our original conclusion regarding PTC- and uORF-containing transcripts. It did change the conclusion that the levels of down-regulated 3'UTR-containing transcripts in *daf-2* mutants were increased by *smg-2(-)* mutations; this actually further strengthened our point that *daf-2* mutations decrease the levels of various NMD target transcripts. Accordingly, we revised our main text as follows,

Results, page 7, line 131: “Next we examined whether the expression of PTC-, uORF-, or long 3'UTR-containing transcripts that were down-regulated in *daf-2* mutants compared to wild-type animals were changed in *smg-2; daf-2* mutants. The expression levels of down-regulated PTC-, uORF-, or 3'UTR-containing transcripts in *daf-2* mutants were increased in *smg-2(-); daf-2(-)* mutants (Fig. 4d-f).”

Why is there a *smg-2* dependence in the down genes, but not in the up genes?

>We appreciate the reviewer's comment. We previously used “Up” transcripts in *daf-2* mutants as controls like we did with “All” transcripts to show that the levels of “Down” transcripts in *daf-2* mutants were increased in *smg-2; daf-2* mutants. Showing both “Up” and “Down” transcripts, one as NMD targets and the other as controls, is a commonly used method for the analysis of NMD target levels using RNA seq. experiments (Bao et al., 2016). As the analysis of *smg-2* dependence of “Up” transcripts in *daf-2* mutants are not meaningful, we removed “Up” genes and now have only the “All” (control) and “Down” (experimental) bar graphs to help readers better understand the data (Fig. 4a,b). In addition, as we stated above, to show the *smg-2* dependency more properly, we changed our analysis using only down-regulated transcripts in *daf-2* mutants compared to wild-type.

5. To complement the RNAi experiments, tissue-specific rescue of *smg-2* should be done.

> We very much appreciate the comments by the reviewer, and have performed the suggested experiments. We generated tissue-specific rescue transgenic worms, *dpy-7p::smg-2::gfp*, *hlh-1p::smg-2::gfp*, *ges-1p::smg-2::gfp*, and *rgef-1p::smg-2::gfp*, for hypodermal, muscle, intestinal, and neuronal expression, respectively. We found that neuron-, hypodermis-, and muscle-specific transgenes partially rescued the longevity of *smg-2(qd101); daf-2(e1370)* (Fig. 6e-g), while the intestine-specific expression of *smg-2* did not (Fig. 6h). Because neurons are the only tissue that showed both requirement and sufficiency of *smg-2* for the longevity of *daf-2* mutants, we modified our Results focusing on the point as follows,

Results, page 8, line 156: "Using tissue-specific rescue experiments, we found that neuronal-, hypodermal-, and muscle-specific *smg-2* expression partially restored longevity in *smg-2(-); daf-2(-)* animals (Fig. 6 e-g), while intestine-specific *smg-2* expression did not (Fig. 6h). Thus, neurons are the only tissue that is required and sufficient for longevity in *daf-2(-)* mutants. In combination with the requirement of neuron-specific *smg-2* activity and the neuronal maintenance of NMD, suggest that RNA surveillance in neurons is particularly crucial for the longevity of *daf-2* mutants."

It would also be nice to determine whether overexpression of *smg-2* can increase lifespan.

We found that *smg-2::gfp* had no effect on the lifespan of wild-type (Fig. 5d). During this revision, we obtained new data showing that overexpression of *smg-1*, which encodes a protein kinase that activates SMG-2 via phosphorylation, increased lifespan (Fig. 5e). This result suggests that activation of SMG-2 by phosphorylation is sufficient for increasing lifespan.

Results, page 7, line 148: "In contrast, overexpression of SMG-1, a protein kinase that phosphorylates and activates SMG-2, increased lifespan (Fig. 5e)."

Reviewer #3 (Expert in NMD; Remarks to the Author):

The authors conclude that while SMG2 deficiency has little effect on WT worms - or a

dietary restriction mimetic, a hif-1 gain-of-function mutant or a mitochondrial respiration-defective mutant - it substantially shortens the long lifespan induced by genetic inhibition of daf2 insulin/IGF-1 receptor. This is true for neurons but not true for intestine-, muscle- or hypodermis-specific SMG2 RNAi, leading the authors to conclude that the maintenance of NMD in neurons is critical for long lifespans of worms with reduced IIS.

While interesting and certainly with important potential, there are problems with this submission, as noted in the itemized review below. Primarily, the authors have not convincingly demonstrated that IIS enhances NMD; if it does, then it seems important to show something about mechanism - how is surveillance enhanced?

> We appreciate the reviewer's comments. We agree with the reviewer that we do not show detailed mechanisms by which reduced IIS enhances NMD. With all due respect, however, we would like to point out that this is the one of the first, if not the first, reports showing that RNA quality control contributes to organismal longevity. Despite rather weak mechanistic data, we believe our functional molecular genetic data are strong enough to serve as an important starting point of a new research field on the role of RNA homeostasis in aging. As described below and above for specific responses, we also included many additional data that strengthened our main conclusions. We hope that the reviewer finds our revised manuscript is greatly improved with these new data and is suitable for publication.

Specific Comments

Abstract, line 8. Here, the authors fail to mention an important aspect of NMD - in the decay of an estimated 10% of normal physiologic transcripts.

> We thank the reviewer for this comment. We now mention this aspect of NMD by changing the Abstract and the Introduction as follows,

Abstract, page 2, line 22: "Here, we show that nonsense-mediated mRNA decay (NMD), which degrades abnormal transcripts as well as **some normal transcripts**, promotes

longevity by enhancing RNA surveillance in *C. elegans*.”

Introduction, page 3, line 48: “NMD also regulates the level of **approximately 10% of endogenous transcripts**, including upstream open reading frames (uORFs)- and long 3’UTR-containing transcripts.”

They also do not discuss how NMD is regulated conditionally by cells to adapt to changing environments, e.g. possibly the one created in response to changes in IGF-1 receptor activity and/or as cells age.

> We thank the reviewer for this insightful comment. We now included the discussion point regarding how NMD is regulated in a tissue-specific manner as follows,

Discussions, page 9, line 180: “One interesting aspect of our study is that the contributions of NMD to longevity in different tissues appear to be different. NMD efficiencies also differ among various mammalian tissues (Huang and Wilkinson, 2012). For example the ratio of PTC-containing *Men1* mRNA to control is variable among 13 different mouse tissues (Zetoune et al., 2008). One possible explanation regarding these tissue-specific effects of NMD is that NMD components are differentially regulated among various tissues by cell type-specific negative-feedback regulation (Huang et al., 2011). Further studies are required to dissect mechanisms by which NMD is tissue-specifically regulated.”

Abstract Line 9-11: "We found that long-lived *daf-2*...mutants displayed enhanced NMD activity...". This is a central argument of the paper, but hasn't been shown convincingly. As far as I can tell, only a "fingerprint" of downregulated (PTC+, uORF-containing but unexpectedly not "long 3'UTR") transcripts found in *daf-2* mutant worms is used to make this conclusion. The reporter from Figure 1 could be used in a *daf-2* background (combined with appropriate half-life studies that show reduction in half-life in the *daf-2* background) to make the argument more convincing.

> This comment is very important. We agree with the reviewer’s comment that mRNA half-life studies will greatly strengthen our point that NMD efficiency is increased in *daf-2* mutants. To our knowledge, however, RNA half-life assay using *C. elegans* is performed with

L1 larval worms treated with α -amanitin (20 $\mu\text{g}/\text{ml}$), as adult worms are resistant to even with 100 $\mu\text{g}/\text{ml}$ of α -amanitin (Miki et al., 2014; Sanford et al., 1983). We wanted to use DRB (5,6-Dichlorobenzimidazole 1- β -D-ribofuranoside) instead of α -amanitin because DRB is faster and more specific to RNA pol II than α -amanitin (Bensaude, 2011; Chen et al., 2008). We found that we needed over 100 $\mu\text{g}/\text{ml}$ concentration of DRB treatment for an effective dose even for L1 larval worms, and it seems very difficult to perform these experiments with adult worms. As we are determining the roles of NMD in adult lifespan, we need to develop and optimize a new mRNA half-life assay, which will take very long and constitute a separate method paper. Therefore, in this manuscript we describe the limitation of our study in the Discussion as follows,

Discussion, page 10, line 196: “One of the limitations of our RNA quantification data is that mRNA levels can be affected both by transcription and by RNA stability. RNA stability can be measured by various approaches, including the measurement of RNA half-life upon chemical inhibition of transcription in a cell culture system (Cheneval et al., 2010). However, the thick cuticle of adult *C. elegans* is not amenable for treatment with transcription inhibitors. In fact, RNA half-life assays in *C. elegans* are conducted in L1 larvae, which lack the tough cuticle (Miki et al., 2014; Sanford et al., 1983), and therefore it will be important to develop and optimize a new mRNA half-life assay for adult *C. elegans* in the future.”

Instead of a half-life assay, we examined the fluorescence levels of the GFP-fused NMD reporter in the *daf-2* mutants, and also used control transgenic worms that do not contain PTC. We normalized the GFP levels of NMD reporter with the control (*lacZ(PTC)/lacZ(WT)*), and found that *daf-2* mutants displayed significantly decreased ratios of GFP intensities in both young (day 1) and old (day 9) worms compared to wild-type (Fig. 4i). We describe these new data as follows,

Results, page 7, line 138: “In addition, *daf-2* mutations decreased the fluorescence intensity ratio of *lacZ(PTC)/lacZ(WT)* GFP reporter, both in young and old animals (Fig. 4i).”

Page 3, line 7. "Ribostasis" is not a word known by this reviewer to be generally used by RNA biologists and is confusing since, a priori, it could be interpreted to mean ribosome homeostasis - "RNAstasis" would be much better, but again is not a term that is used. This reviewer recommends simply stating "RNA homeostasis", which everyone will understand.

> We thank and agree with the reviewer's comments. We changed "Ribostasis" to "RNA homeostasis", and revised our manuscript as follows,

Introduction, page 3, line 49: "Therefore, NMD acts as a crucial regulator of general **RNA homeostasis**, and prevents accumulation of potentially deleterious non-functional proteins."

Results, page 3, line 56: "To determine whether NMD-mediated **RNA homeostasis** is important for aging and longevity regulation, we first examined the level of NMD activity using a reporter."

Results, page 6, line 117: "As NMD plays an important role in RNA surveillance, we examined whether the enhanced **RNA homeostasis** by NMD underlies the longevity of *daf-2* mutants."

Discussion, page 10, line 204: "We report that IIS, one of the most evolutionarily conserved aging-regulatory signaling pathways, regulates RNA surveillance through NMD and **RNA homeostasis** to influence aging."

Discussion, page 10, line 212: "As humans live much longer than animals with comparable sizes and metabolic rates, and NMD components are well conserved, it is likely that humans are equipped with protective systems for **RNA homeostasis** as well."

Figure 1 and page 3, line 16 of conclusion. How much of the increase in mRNA abundance when SMG2 is downregulated is transcriptional vs. due to an increase in mRNA half-life? Until this is known, the authors cannot comment on the efficiency of NMD. Also related: The natural decline in NMD activity with age would seem to be a central argument for relevance of the story. The only real evidence for this is Figure 1d, but the data are confounded by the possibility that the activity of the Sec23p promoter used to drive the reporter changes with

age. Half-life studies can resolve this.

> We very much appreciate these excellent points of the reviewer. Although we could not perform a half-life assay, we used control (*sec-23p::gfp::lacZ*) transgenic worms that do not harbor PTC, and used them for the normalization of the GFP intensity of the NMD reporter (*sec-23p::gfp::lacZ(PTC)*). We found that *sec-23* promoter activity did not change during aging in control worms, and after normalization we found that the ratio of *lacZ(PTC)/lacZ(WT)* GFP intensity increased with age, and this is consistent with our original conclusion. We now describe this result as follows,

Results, page 4, line 63: “Importantly, aged (Day 9 adult) worms displayed an increased ratio of *lacZ(PTC)/lacZ(WT)* GFP intensity, indicating an age-dependent impairment of NMD activity (Fig. 1b-d).”

To further support our conclusion, we also performed RT-PCR to detect endogenous PTC-positive (*rpl-12(PTC)*) and negative (*rpl-12(WT)*) transcripts simultaneously using day 1 (young) and day 9 (old) worms. We found that the ratio of *rpl-12(PTC)/rpl-12(WT)* transcripts was increased with age. We describe this result in the text as follows,

Results, page 4, line 64: “Next, we examined whether the mRNA levels of PTC-containing *rpl-12 (rpl-12(PTC))*, a well-established endogenous target of NMD (Supplementary Fig. 1e) (Mitrovich and Anderson, 2000), changed during aging. We found that old animals displayed increased *rpl-12(PTC)* levels when they were normalized with *rpl-12(WT)* transcripts that do not contain PTC, compared to that in young worms (Fig. 1e). This result supports the idea that NMD activity decreases during aging.”

We also generated tissue-specific promoter-driven NMD reporters (neurons, hypodermis, muscles, and intestine) with control transgenic worms with the corresponding promoters. We found that the normalized GFP intensity of *lacZ(PTC)/lacZ(WT)* reporter levels were increased during aging in the hypodermis, muscles, and intestine, while those in neurons were not significantly changed (Fig. 1f). We added these data and revised the manuscript as follows,

Results, page 4, line 70: “We sought to determine in which tissues NMD activity changed during aging by examining neuron-, hypodermis-, muscle-, and intestine-specific NMD

reporters. We found that the normalized GFP intensities of *lacZ(PTC)/lacZ(WT)* were increased during aging in the hypodermis, muscle, and intestine, while those in neurons were largely unaffected with age (Fig 1f). This result raises a possibility that the maintenance of NMD activity in neurons is sustained longer than that in other tissues.”

Figures 1 and 2. Since *C. elegans* SMG2, which is orthologous to human/mammalian UPF1, is used in many pathways other than NMD, at least in human/mammalian cells, the authors would do well to vary the abundance of another NMD factor. In fact, the authors later show that shorter life-spans typify worms that lack SMG-2 relative to worms lacking one of several other NMD factors (Figure 2g). How does the efficiency of NMD compare in worms deficient in the various NMD factors? This is a key issue that needs to be resolved before the authors can conclude what they do.

> This is a very critical point. To examine the efficiency of NMD components, we measured the GFP intensity of the NMD reporter upon knocking down each of *smg-1* through *smg-7*. We found that RNAi targeting *smg-1* through *smg-5* increased GFP intensity, whereas *smg-6* and *smg-7* RNAi did not (Fig. 3a). We also noticed that the extent of the increase in fluorescence level was variable among *smg-1* through *smg-5* RNAi conditions. Because of these results, we focused our lifespan analysis description on *smg-1* through *smg-5*. We updated our manuscript as follows,

Results, page 6, line 107: “The NMD pathway is regulated by the NMD complex, composed of many components (SMG-1 through SMG-7). We found that knockdown of *smg-1* through *smg-5* significantly increased the GFP intensity of the NMD reporter, but *smg-6* and *smg-7* RNAi did not (Fig. 3a). We also noticed that the effects of *smg-1* through *smg-5* RNAi on the GFP intensity were variable (Fig. 3a). These data are consistent with reports showing that NMD components play both overlapping and distinct roles, and their effects on a same target can be variable (Johns et al., 2007). We also found that RNAi knockdown of each of *smg-1*, -2, -3, and -5, significantly suppressed the long lifespan of *daf-2* mutants (Fig. 3b-d,f); *smg-4* RNAi decreased the long lifespan of *daf-2* mutants in one out of two trials (Fig. 3e). Thus, NMD appears to promote longevity in animals with reduced insulin/IGF-1 signaling.”

Although we obtained negative data using *smg-6* RNAi and *smg-7* RNAi for the NMD reporter, we still included our lifespan results using *smg-6* and *smg-7* RNAi in the Supplementary Figure 4 and discussed the negative data in the legends.

Supplementary Fig. 4a legend, page 8, line 2: “*smg-6* RNAi shortened the lifespan of *daf-2* mutants in 1 out of 2 trials.”

Supplementary Fig. 4b legend, page 8, line 3: “*smg-7* RNAi did not affect the lifespan of wild-type or *daf-2* mutant animals. Because *smg-7* RNAi did not affect the GFP intensity of NMD reporter (Fig. 3a) or lifespan, we cannot exclude the possibility that *smg-7* RNAi treatment did not effectively knock down *smg-7* gene.”

Page 5, line 3. Why do *daf-2* mutations reduce the level of mRNAs with an uORF or PTC but not "long" (what do the authors mean by "long"?) 3'UTR?

> We really appreciate the reviewer’s comment. As the reviewer pointed out, we did not define the length for long 3’UTR specifically and agree with the reviewer that our analysis was ambiguous. In this revised manuscript, we divided the lengths of 3’UTRs into 3 groups, ≥ 350 , $350 < < 1500$, ≥ 1500 , for short, medium, and long 3’UTRs, respectively, following a previous report (Fanourgakis and Lesche, 2016). We then analyzed the fold changes for these categories of transcripts. As expected, we found that the levels of long 3’UTR-containing transcripts were increased in *smg-2* mutants (Supplementary Fig. 8e). We also found that *daf-2* mutations decreased the level of long 3’UTR containing transcripts (Fig. 4c). In addition, the decreased levels of long 3’UTR-containing transcripts in *daf-2* compared to wild-type were increased by *smg-2* mutations (Fig. 4f). Our previous analysis, which showed the fold changes of all transcripts in *daf-2* mutants compared to wild-type, may have hindered the *smg-2* dependency on down-regulated and long 3’UTR-containing transcripts in *daf-2* mutants. We describe this new analysis as follows,

Results, page 7, line 131: “Next we examined whether the expression of PTC-, uORF-, or long 3’UTR-containing transcripts that were down-regulated in *daf-2* mutants compared to wild-type animals were changed in *smg-2*; *daf-2* mutants. The expression levels of down-regulated PTC-, uORF-, or 3’UTR-containing transcripts in *daf-2* mutants were increased in *smg-2(-)*; *daf-2(-)* mutants (Fig. 4d-f).”

Fig. 4c legend, page 28, line 654: “Box plots showing the fold changes in *daf-2(-)* mutants compared to wild-type for the transcripts containing long, medium, or short 3’UTR lengths. Short (≤ 350 nt), medium ($350 \text{ nt} < < 1500$ nt), and long (≥ 1500 nt). 3’UTR lengths were defined following a previous report (Fanourgakis and Lesche, 2016). Thick black lines indicate median values (** $p < 0.01$, *** $p < 0.001$, Wilcoxon rank sum test).”

-The suggestion that *daf-2* mutations promote NMD is not particularly solid - can the authors reproduce this finding using various types of NMD reporters?

> This is a great suggestion. As we stated above, we found that the GFP-fused NMD reporter level was decreased in *daf-2* mutants, but it would be much better if we had the data using various types of NMD reporters. However, to our knowledge the NMD reporter that we used is the only available one in *C. elegans* system. Generating and optimizing new reporter systems would need much more time than the allowed revision time. Instead, we showed changes in mRNA levels of an endogenous NMD target *rpl-12* using qRT-PCR.

Results, page 7, line 136: “We found that the mRNA level of PTC-containing *rpl-12* measured by using qRT-PCR was decreased in *daf-2* mutants, in a *smg-2*-dependent manner (Fig. 4g,h).”

In addition, we also qualified our claim that *daf-2* mutations increase the NMD activity as follows,

From “We propose that reduced insulin/IGF-1 signaling increases lifespan through up-regulation of NMD activity...”

=> To Abstract, page 2, line 31: “We propose that up-regulation of NMD activity contributes to long lifespan caused by reduced insulin/IGF-1 signaling...”

From “Together, these data suggest that *daf-2* mutations promote NMD activity, which should contribute to longevity by reducing the levels of target transcripts, including uORF-containing mRNAs and PTC-positive mRNAs.”

=> To Results, page 7, line 139: “Together, these data raise the possibility that decreased levels of PTC-, uORF-, or long-3’UTR-containing transcripts in *daf-2* mutants may

contribute to longevity.”

Page 6. Does IIS, i.e. insulin/IGF (insulin-like growth factor)-like signaling, really regulate RNA surveillance through NMD? This reviewer is not convinced.

> We thank the reviewer for raising this issue. Although our RNA sequencing analysis and NMD reporter data support the idea that *daf-2* mutants may have enhanced NMD activity, we agree with the reviewer’s comments that we should perform a half-life assay and use more NMD reporters to show that NMD activity is indeed enhanced in *daf-2* mutants in future studies. Thus, we revised our manuscript to tone down our findings as above.

We also added a discussion point regarding a possible mechanism by which NMD is regulated via insulin/IGF-1 signaling.

Discussion, page 9, line 187: “An important remaining question is then how *daf-2*/insulin/IGF-1 receptor mutations affect NMD activity in *C. elegans*. Ca^{2+} acts as a robust NMD regulator (Nickless et al., 2014), and insulin signaling and Ca^{2+} homeostasis is tightly linked, as proper regulation of intracellular Ca^{2+} level is required for insulin secretion (Draznin, 1988). Insulin signaling also affects intracellular Ca^{2+} level, as overexpression of insulin receptor or insulin receptor substrate-1 increases cytosolic Ca^{2+} level (Borge et al., 2002). In addition, insulin is known as an activator of the ryanodine receptor and the inositol 1,4,5-triphosphate receptor, which increases Ca^{2+} levels in the cytosol (Contreras-Ferrat et al., 2014). As intracellular Ca^{2+} is known to decrease NMD activity (Nickless et al., 2014), we speculate that one possibility is that *daf-2* mutations may decrease cytosolic Ca^{2+} levels and up-regulate NMD perhaps via a similar mechanism.”

We hope these text changes will be satisfactory to the reviewer.

Minor Comment

Abstract, line 7. RNA is the second component in the flow of information (Central Dogma):

DNA -> mRNA -> protein

We thank the reviewer for pointing this out. We changed the text as follows,

Introduction, page 3, line 42: “However, the role of homeostatic regulation of the second component of the Central Dogma, RNA, in aging is unknown.”

References

- Bao, J., Vitting-Seerup, K., Waage, J., Tang, C., Ge, Y., Porse, B.T., and Yan, W. (2016). UPF2-Dependent Nonsense-Mediated mRNA Decay Pathway Is Essential for Spermatogenesis by Selectively Eliminating Longer 3'UTR Transcripts. *PLoS genetics* *12*, e1005863.
- Bensaude, O. (2011). Inhibiting eukaryotic transcription: Which compound to choose? How to evaluate its activity? *Transcription* *2*, 103-108.
- Borge, P.D., Moibi, J., Greene, S.R., Trucco, M., Young, R.A., Gao, Z., and Wolf, B.A. (2002). Insulin receptor signaling and sarco/endoplasmic reticulum calcium ATPase in beta-cells. *Diabetes* *51 Suppl 3*, S427-433.
- Chen, C.Y., Ezzeddine, N., and Shyu, A.B. (2008). Messenger RNA half-life measurements in mammalian cells. *Methods in enzymology* *448*, 335-357.
- Cheneval, D., Kastelic, T., Fuerst, P., and Parker, C.N. (2010). A review of methods to monitor the modulation of mRNA stability: a novel approach to drug discovery and therapeutic intervention. *Journal of biomolecular screening* *15*, 609-622.
- Contreras-Ferrat, A., Lavandero, S., Jaimovich, E., and Klip, A. (2014). Calcium signaling in insulin action on striated muscle. *Cell calcium* *56*, 390-396.
- Draznin, B. (1988). Intracellular calcium, insulin secretion, and action. *The American journal of medicine* *85*, 44-58.
- Fanourgakis, G., and Lesche, M. (2016). Chromatoid Body Protein TDRD6 Supports Long 3' UTR Triggered Nonsense Mediated mRNA Decay. *PLoS genetics* *12*, e1005857.
- Hsin, H., and Kenyon, C. (1999). Signals from the reproductive system regulate the lifespan of *C. elegans*. *Nature* *399*, 362-366.
- Hu, P.J. (2007). Dauer. *WormBook : the online review of C elegans biology*, 1-19.
- Huang, L., Lou, C.H., Chan, W., Shum, E.Y., Shao, A., Stone, E., Karam, R., Song, H.W., and

- Wilkinson, M.F. (2011). RNA homeostasis governed by cell type-specific and branched feedback loops acting on NMD. *Molecular cell* 43, 950-961.
- Huang, L., and Wilkinson, M.F. (2012). Regulation of nonsense-mediated mRNA decay. *Wiley interdisciplinary reviews RNA* 3, 807-828.
- Johns, L., Grimson, A., Kuchma, S.L., Newman, C.L., and Anderson, P. (2007). *Caenorhabditis elegans* SMG-2 selectively marks mRNAs containing premature translation termination codons. *Molecular and cellular biology* 27, 5630-5638.
- Kaletsky, R., Lakhina, V., Arey, R., Williams, A., Landis, J., Ashraf, J., and Murphy, C.T. (2015). The *C. elegans* adult neuronal IIS/FOXO transcriptome reveals adult phenotype regulators. *Nature*.
- Kamath, R.S., Martinez-Campos, M., Zipperlen, P., Fraser, A.G., and Ahringer, J. (2001). Effectiveness of specific RNA-mediated interference through ingested double-stranded RNA in *Caenorhabditis elegans*. *Genome biology* 2, Research0002.
- Kenyon, C.J. (2010). The genetics of ageing. *Nature* 464, 504-512.
- Miki, T.S., Ruegger, S., Gaidatzis, D., Stadler, M.B., and Grosshans, H. (2014). Engineering of a conditional allele reveals multiple roles of XRN2 in *Caenorhabditis elegans* development and substrate specificity in microRNA turnover. *Nucleic acids research* 42, 4056-4067.
- Mitrovich, Q.M., and Anderson, P. (2000). Unproductively spliced ribosomal protein mRNAs are natural targets of mRNA surveillance in *C. elegans*. *Genes & development* 14, 2173-2184.
- Nickless, A., Jackson, E., Marasa, J., Nugent, P., Mercer, R.W., Piwnica-Worms, D., and You, Z. (2014). Intracellular calcium regulates nonsense-mediated mRNA decay. *Nature medicine* 20, 961-966.
- Sanford, T., Golomb, M., and Riddle, D.L. (1983). RNA polymerase II from wild type and alpha-amanitin-resistant strains of *Caenorhabditis elegans*. *The Journal of biological chemistry* 258, 12804-12809.
- Timmons, L., Court, D.L., and Fire, A. (2001). Ingestion of bacterially expressed dsRNAs can produce specific and potent genetic interference in *Caenorhabditis elegans*. *Gene* 263, 103-112.
- Wang, D., and Ruvkun, G. (2004). Regulation of *Caenorhabditis elegans* RNA interference

by the *daf-2* insulin stress and longevity signaling pathway. Cold Spring Harbor symposia on quantitative biology 69, 429-431.

Zetoune, A.B., Fontaniere, S., Magnin, D., Anczukow, O., Buisson, M., Zhang, C.X., and Mazoyer, S. (2008). Comparison of nonsense-mediated mRNA decay efficiency in various murine tissues. BMC genetics 9, 83.

Reviewer #1 (Remarks to the Author)

The findings described in this study are worthy of publication in Nature Communications. After addressing most of the reviewers' comments, the paper has significantly improved and the conclusions are more robust.

After these changes, I strongly recommend to publish it.

I only suggest two minor points that need to be further addressed:

1- In page 5, line 88, the authors mentioned: "The specific longevity-suppressing effect of smg-2 RNAi on daf-2 mutants is likely because daf-2 mutants are RNAi sensitive^{13, 14}." Although this is an interesting hypothesis supported by the available literature, I'm not convinced this explains the results obtained by the authors and it might be other reasons. I suggest that the authors discuss other possibilities. Otherwise, the authors should strengthen their current hypothesis with glp-1, eat-2 (etc) mutants in an RRF-1 mutant background.

2- Fig. 3a: Did the authors checked the knockdown levels induced by the different smg RNAs. A lack of effective knockdown could explain why they didn't observed changes induced by smg-6 and smg-7 knockdown.

Reviewer #2 (Remarks to the Author)

The manuscript states that it is the first demonstration of RNA homeostasis and longevity. First, there is essentially no effect on longevity, except in certain mutant backgrounds (see below). Second, there is lots of work suggesting a role for RNA in aging, including many neurodegenerative diseases, but also microRNA, long non-coding RNA, etc. Rather than attempting to minimize this previous work, it would be better to place the current effort in the appropriate context. Third, this manuscript does not analyze homeostasis, there are no experiments suggesting that this is a homeostatic process.

The revised paper clearly shows that under normal circumstances, NMD genes do not affect lifespan. That is one of the strongest conclusions of the paper, particularly coupled with the finding that NMD of the only endogenous target tested is barely affected in old animals. The NMD reporter does show a large difference in old animals.

In the daf-2 background, the story is different. Here, smg-2 reduces lifespan of daf-2, and also reduces lifespan of other aging mutants. Further, the data are quite strong that daf-2 increases activity of NMD. This an interesting finding. But there is a complete lack of mechanism. One mechanism, that smg-2 is upregulated in daf-2, is tested but found to be negative. A second mechanism, that smg affects daf-16, is tested, but found to act in the opposite direction than expected. But surprisingly, especially considering the authors' extensive work on this pathway, the potential function of daf-16, the best-known mediator of daf-2, in mediating the increase in NMD in daf-2 mutants is not addressed. Also, sequencing data in neurons from smg/daf-2 mutants could reveal some mechanisms.

The findings on the effect of the smg genes on extended longevity of some aging mutants is certainly interesting, but a clearer statement of the findings, and a better attempt to explore mechanism, would enhance the findings. Also, in light of the discrepancy between rna and null mutant phenotypes, the manuscript should make less emphasis on rna.

Reviewer #3 (Remarks to the Author)

It is difficult for me to be supportive of this submission since I disagree with the authors' conclusions that mechanism isn't necessary. That was one of my concerns that I stated in my review, but the authors feel it is not important: how does reduced insulin/IGF-1 signaling via Daf-2 mutations increase the efficiency of NMD and thereby lifespan? Also, they still haven't shown that mRNA half-life, i.e. NMD per se, is affected - they used one reporter with and without a PTC, arguing that the promoter is the same. However, that promoter may be immune to the siRNAs they use, while others are not.

Reviewer #1 (Remarks to the Author):

The findings described in this study are worthy of publication in Nature Communications. After addressing most of the reviewers' comments, the paper has significantly improved and the conclusions are more robust.

After these changes, I strongly recommend to publish it.

I only suggest two minor points that need to be further addressed:

Minor points:

- In page 5, line 88, the authors mentioned: "The specific longevity-suppressing effect of *smg-2* RNAi on *daf-2* mutants is likely because *daf-2* mutants are RNAi sensitive^{13, 14}." Although this is an interesting hypothesis supported by the available literature, I'm not convinced this explains the results obtained by the authors and it might be other reasons. I suggest that the authors discuss other possibilities. Otherwise, the authors should strengthen their current hypothesis with *glp-1*, *eat-2* (etc) mutants in an RRF-1 mutant background.

> We thank the reviewer for raising this issue. We agree with the reviewer's point that there may be other reasons why *smg-2* RNAi decreased the longevity of *daf-2* mutants but not that of other mutants. In our revised manuscript, we removed our description regarding the effects of the *smg-2* RNAi on *daf-2*(-)-longevity in the main text to focus more on the data obtained by using *smg-2* mutants; this is also based on a comment by reviewer 2 (see below). We also removed the sentence regarding the RNAi sensitivity of *daf-2* mutants. That is because we did not observe enhanced RNAi efficiency in *daf-2* mutants compared to wild-type, when we performed qRT-PCR to test the RNAi efficiencies of *smg* gene RNAi clones (Supplementary Fig. 6). Thus, we discuss why *smg-2* RNAi treatment decreased the longevity of *daf-2* mutants but not that of other longevity mutants with another possibility in the Supplementary figure legends as follows,

Figure 2 legends, page 29, line 694: "Please see Supplementary Fig. 2c-f for data regarding the effects of *smg-2* RNAi on the longevity of *isp-1*, *eat-2*, *glp-1*, and *vhl-1* mutants."

Supplementary Fig. 2 legends, page 4: “Specific lifespan-decreasing effects of *smg-2* RNAi on *daf-2* mutants may be due to variability in RNAi efficiency, as we did not perform all the RNAi-based lifespan assays using various longevity mutants in the same experimental set.”

- Fig. 3a: Did the authors checked the knockdown levels induced by the different *smg* RNA is. A lack of effective knockdown could explain why they didn't observed changes induced by *smg-6* and *smg-7* knockdown.

> We appreciate the reviewer's comment, and examined the mRNA levels of *smg-1* through *smg-7* after knocking down each of these seven *smg* genes by using RNAi. We found that mRNA levels of *smg-1* through *smg-5* mRNA levels were decreased after treatment with respective RNAi clones in wild-type and *daf-2* mutant backgrounds. In contrast, *smg-6* or *smg-7* transcript levels were not significantly affected by treatment with respective RNAi. Thus, we removed our previous data regarding GFP-fused NMD reporters and lifespan assays using *smg-6* RNAi and *smg-7* RNAi. Instead, we included the data showing mRNA levels of these *smg* genes, and changed our manuscript as follows,

Results, page 6, line 111: “We first determined the efficiency of RNAi clones targeting *smg-1* through *smg-7*. We found that RNAi clones targeting *smg-1* through *smg-5* decreased the mRNA levels of respective *smg* genes in wild-type and *daf-2* mutant backgrounds, whereas *smg-6* RNAi or *smg-7* RNAi clones did not (Supplementary Fig. 6).”

Reviewer #2 (Remarks to the Author):

The manuscript states that it is the first demonstration of RNA homeostasis and longevity. First, there is essentially no effect on longevity, except in certain mutant backgrounds (see below).

Second, there is lots of work suggesting a role for RNA in aging, including many neurodegenerative diseases, but also microRNA, long non-coding RNA, etc. Rather than attempting to minimize this previous work, it would be better to place the current effort in the appropriate context.

> We sincerely appreciate this reviewer's comment, and agree with this point. Accordingly we moderated our claim by focusing on RNA quality control and NMD, and modified our text in several places as follows,

Introduction, page 3, line 42: "Regarding RNA, several neurodegenerative disorders are associated with defects in RNA-binding protein function (Arai et al., 2006; Neumann et al., 2006), and many non-coding RNAs such as microRNAs and long non-coding RNAs play regulatory roles in longevity (Dluzen et al., 2016; Inukai and Slack, 2013; Kour and Rath, 2016). However, whether RNA quality control affects aging is largely unknown."

Third, this manuscript does not analyze homeostasis, there are no experiments suggesting that this is a homeostatic process.

> We thank the reviewer for this valuable comment, and agree with the reviewer that we did not show nonsense-mediated mRNA decay (NMD) regulated lifespan through RNA homeostasis. We therefore removed "RNA homeostasis" throughout text and used "RNA surveillance", "RNA quality control" or simply "nonsense-mediated mRNA decay" as follows,

Introduction, page 2, line 31: "NMD in the nervous system of the animals was particularly important for RNA quality control to promote longevity."

Introduction, page 3, line 50: "Therefore, NMD acts as a crucial regulator of general RNA quality control, and prevents accumulation of potentially deleterious non-functional proteins."

Results, page 4, line 59: "To determine whether NMD-mediated RNA surveillance is important for aging and longevity regulation, we first examined the level of NMD activity

using a reporter.”

Results, page 6, line 124: “As NMD plays an important role in RNA surveillance, we examined whether the enhanced **NMD** underlies the longevity of *daf-2* mutants.”

Discussion, page 10, line 204: “Here, our data suggest that NMD-mediated **RNA quality control** is critical for longevity in *C. elegans*.”

Discussion, page 10, line 210: “Thus neurons may need to maintain better **RNA quality control** via enhanced NMD.”

Discussion, page 11, line 229: “We report that IIS, one of the most evolutionarily conserved aging-regulatory signaling pathways, regulates **RNA surveillance** through NMD to influence aging.”

Discussion, page 11, line 232: “Our findings suggest that IIS actively regulates **quality control systems** for all three components of the Central Dogma (DNA, RNA and proteins), and this may at least in part underlie the extreme longevity of *daf-2* mutants.”

Discussion, page 11, line 237: “As humans live much longer than animals with comparable sizes and metabolic rates, and NMD components are well conserved, it is likely that humans are equipped with protective systems for **RNA** as well”

The revised paper clearly shows that under normal circumstances, NMD genes do not affect lifespan. That is one of the strongest conclusions of the paper, particularly coupled with the finding that NMD of the only endogenous target tested is barely affected in old animals. The NMD reporter does show a large difference in old animals. In the *daf-2* background, the story is different. Here, *smg-2* reduces lifespan of *daf-2*, and also reduces lifespan of other aging mutants. Further, the data are quite strong that *daf-2* increases activity of NMD. This an interesting finding. But there is a complete lack of mechanism. One mechanism, that *smg-2* is upregulated in *daf-2*, is tested but found to be negative. A second mechanism, that *smg* affects *daf-16*, is tested, but found to act in the opposite direction than expected. But surprisingly, especially considering the authors' extensive work on this pathway, the potential function of *daf-16*, the best-known mediator of *daf-2*, in mediating the increase in NMD in *daf-2* mutants

is not addressed.

> We appreciate this reviewer's comment. As mechanistic data, we identified and characterized a functionally important target of SMG-2, *yars-2*/tRNA synthetase, which contributes to the longevity of *daf-2* mutants. We describe these results in the next paragraph.

Also, sequencing data in neurons from *smg/daf-2* mutants could reveal some mechanisms. The findings on the effect of the *smg* genes on extended longevity of some aging mutants is certainly interesting, but a clearer statement of the findings, and a better attempt to explore mechanism, would enhance the findings.

> We thank the reviewer for this insightful comment. During the course of our current second revision, we identified a functionally important target of NMD that confers the longevity of *daf-2* mutants. Based on the gene ontology analysis of our mRNA seq data, we found that the transcript levels of *yars-2*/tRNA synthetase were decreased in *daf-2* mutants in a *smg-2*-dependent manner (Figure 7a,b). We therefore tested whether the changes in *yars-2* transcript levels contributed to *daf-2*(-) longevity. We found that knockdown of *yars-2* completely restored longevity in *smg-2; daf-2* mutants (Figure 7c), while it had no effect on the lifespan of wild-type, *smg-2* or *daf-2* mutants (Supplementary Fig. 13c,d). In addition, we found that the levels of *yars-2* were decreased in other longevity mutants, including *isp-1* and *glp-1* mutants (Supplementary Fig. 13h,i). These data indicate that decreases in the mRNA levels of *yars-2*/tRNA synthetase via NMD contribute to the longevity of *daf-2* mutants and possibly other longevity mutants, including *isp-1* and *glp-1* mutants. We now describe these findings in the text as follows,

Results, page 9, line 186: "Next, we sought to identify targets of NMD that mediated the long lifespan of *daf-2* mutants. Because the main role of NMD is degradation of its target mRNAs, we focused on mRNAs that were down-regulated in *daf-2* mutants. We performed gene ontology (GO) analysis with PTC- or uORF-containing transcripts, which are potential NMD targets, and whose levels are decreased in *daf-2* mutants in a *smg-2*-dependent manner. We found that a GO term regarding tyrosyl-tRNA aminoacylation, which includes *yars-1* and *yars-2*, was highly enriched (Figure 7a, Supplementary Fig. 12c); *yars-1* and *yars-2* encode cytosolic and mitochondrial tyrosyl-tRNA synthetases,

respectively. We confirmed that mRNA levels of *yars-2b.1*, one of the isoforms of *yars-2* and an NMD target, were significantly decreased in *daf-2* mutants in a *smg-2*-dependent manner (Figure 7b, Supplementary Fig. 13a,b). We then tested whether changes in *yars-2* mRNA levels by *smg-2* contributed to the longevity of *daf-2* mutants, by performing lifespan assays upon knocking down *yars-2*. We found that *yars-2* RNAi completely restored longevity in *smg-2; daf-2* mutants (Figure 7c), while having no effect on the lifespan of wild-type, *smg-2* or *daf-2* mutants (Supplementary Fig. 13c,d). These data suggest that an NMD-dependent decrease in *yars-2*/tRNA synthetase mRNA levels is crucial for the longevity of *daf-2* mutants.”

Figure 7 legends, page 32, line 763: “The mRNA levels of *yars-2a* isoform, which is not a target of NMD, were not significantly changed by *daf-2* mutations (Supplementary Fig. 13e,f).”

Figure 7 legends, page 32, line 765: “The mRNA levels of *yars-1* were not changed (Supplementary Fig. 13g).”

Figure 7 legends, page 32, line 766: “The mRNA levels of *yars-2b.1* were decreased in long lived *isp-1* and *glp-1* mutants (Supplementary Fig. 13h,i).”

Also, in light of the discrepancy between *rnai* and null mutant phenotypes, the manuscript should make less emphasis on *rnai*.

> We thank the reviewer for this comment. We now removed the description regarding lifespan data using *smg-2* RNAi in the main text, and instead included the description in figure legends and supplementary figure legends as follows,

Figure 2 legends, page 29, line 694: “Please see Supplementary Fig. 2c-f for data regarding the effects of *smg-2* RNAi on the longevity of *isp-1*, *eat-2*, *glp-1*, and *vhl-1* mutants.”

Supplementary Fig. 2 legends, page 4: “(c-f) The long lifespan induced by *isp-1(qm150)* (*isp-1(-)*) (c), *eat-2(ad1116)* (*eat-2(-)*) (d), *glp-1(e2141)* (*glp-1(-)*) (e), or *vhl-1(ok161)* (*vhl-1(-)*) (f) mutations was not changed by *smg-2* RNAi treatment. Specific lifespan-

decreasing effects of *smg-2* RNAi on *daf-2* mutants may be due to variability in RNAi efficiency, as we did not perform all the RNAi-based lifespan assays using various longevity mutants in the same experimental set.”

Reviewer #3 (Remarks to the Author):

It is difficult for me to be supportive of this submission since I disagree with the authors' conclusions that mechanism isn't necessary. That was one of my concerns that I stated in my review, but the authors feel it is not important: how does reduced insulin/IGF-1 signaling via *Daf-2* mutations increase the efficiency of NMD and thereby lifespan?

> We totally agree with the reviewer's point that the question of how insulin/IGF-1 signaling increases NMD and thereby lifespan is a very important issue, and therefore we further investigated. In our revised manuscript we have identified a functionally crucial *daf-2*-dependent NMD target, *yars-2*/tRNA synthetase, which contributes to long lifespan. From gene ontology analysis using potential NMD targets whose levels were decreased in *daf-2* mutants in a *smg-2*-dependent manner, we found that tyrosyl-tRNA aminoacylation term was enriched (Figure 7a); these include *yars-1* and *yars-2*. We found that knockdown of *yars-2* fully restored longevity in *smg-2; daf-2* double mutants (Figure 7c). In addition, the mRNA levels of *yars-2* were decreased in other longevity mutants, *isp-1* and *glp-1* mutants (Supplementary Fig. 13h,i). Thus, reduced mRNA levels of *yars-2* via NMD appear to contribute to the longevity of *daf-2(-)* as well as other longevity mutants. With these new findings we modified our manuscript as follows,

Results, page 9, line 186: “Next, we sought to identify targets of NMD that mediated the long lifespan of *daf-2* mutants. Because the main role of NMD is degradation of its target mRNAs, we focused on mRNAs that were down-regulated in *daf-2* mutants. We performed gene ontology (GO) analysis with PTC- or uORF-containing transcripts, which are potential NMD targets, and whose levels are decreased in *daf-2* mutants in a *smg-2*-dependent manner. We found that a GO term regarding tyrosyl-tRNA aminoacylation, which includes *yars-1* and *yars-2*, was highly enriched (Figure 7a, Supplementary Fig. 12c); *yars-1* and *yars-2* encode cytosolic and mitochondrial tyrosyl-tRNA synthetases, respectively. We confirmed that mRNA levels of *yars-2b.1*, one of the isoforms of *yars-2*

and an NMD target, were significantly decreased in *daf-2* mutants in a *smg-2*-dependent manner (Figure 7b, Supplementary Fig. 13a,b). We then tested whether changes in *yars-2* mRNA levels by *smg-2* contributed to the longevity of *daf-2* mutants, by performing lifespan assays upon knocking down *yars-2*. We found that *yars-2* RNAi completely restored longevity in *smg-2; daf-2* mutants (Figure 7c), while having no effect on the lifespan of wild-type, *smg-2* or *daf-2* mutants (Supplementary Fig. 13c,d). These data suggest that an NMD-dependent decrease in *yars-2*/tRNA synthetase mRNA levels is crucial for the longevity of *daf-2* mutants.”

Figure 7 legends, page 32, line 763: “The mRNA levels of *yars-2a* isoform, which is not a target of NMD, were not significantly changed by *daf-2* mutations (Supplementary Fig. 13e,f).”

Figure 7 legends, page 32, line 765: “The mRNA levels of *yars-1* were not changed (Supplementary Fig. 13g).”

Figure 7 legends, page 32, line 766: “The mRNA levels of *yars-2b.1* were decreased in long lived *isp-1* and *glp-1* mutants (Supplementary Fig. 13h,i).”

Also, they still haven't shown that mRNA half-life, i.e. NMD per se, is affected - they used one reporter with and without a PTC, arguing that the promoter is the same. However, that promoter may be immune to the siRNAs they use, while others are not.

> We appreciate the reviewer's comment. First, we would like to mention that we have used a total of five different promoter-driven NMD reporters (Figure 1b,c,f), and we used mutants rather than RNAi for genetic intervention (Figure 4i). Thus, we believe that the results from experiments using our NMD reporters are valid.

Nevertheless, we totally agree with the reviewer that it is extremely important for us to measure mRNA half-life. Therefore, for the last four months we have made substantial efforts to establish mRNA half-life assays in *C. elegans*; using these assays, we obtained results consistent with our initial conclusions. We first optimized the condition for RNA half-life assays using adult animals; to our knowledge this is the first time such an assay has been done in adult *C. elegans*. Importantly, we found that RNA half-lives of selected NMD targets

were reduced in *daf-2* mutants compared to wild-type (Figure 4j,k). We found that the effect of *daf-2* mutations on RNA half-lives was *smg-2* dependent (Figure 4j,k). These results suggest that NMD is enhanced in *daf-2* mutants. We describe these findings as follows,

Results, page 8, line 148: “We also found that RNA half-lives of selected NMD targets were reduced by *daf-2* mutations (Figure 4j,k). Moreover, the effect of *daf-2* mutations on RNA half-lives was dependent on *smg-2* (Figure 4j,k). Overall these data suggest that NMD is enhanced in *daf-2* mutants.”

In addition, we describe how we optimized experimental conditions for our RNA half-life assays using adult animals in the Supplementary figure legends:

Supplementary Fig. 10 legends, page 20: “We treated worms with DRB, one of established transcription inhibitors used for measuring mRNA half-life (Bensaude, 2011; Chen et al., 2008), to block transcription. As DRB is dissolved in DMSO, which can be toxic to worms (Solis and Petrascheck, 2011), we first tested the effects of various concentrations of DMSO (1-5%) on the growth of worms. We found that 1% DMSO was the highest concentration that did not delay the development of *C. elegans* (Supplementary Fig. 10a). Next we tried to titrate optimal concentrations of DRB that inhibited growth, which reflects inhibited transcription (Miki et al., 2014), in wild-type, *daf-2*, *smg-2*, and *smg-2*; *daf-2* mutants. We found that over 200 µg/ml of DRB treatment was sufficient for inhibiting the growth (Supplementary Fig. 10b). Next, we measured the mRNA levels of *pre-eft-3*, (a positive control) by using qRT-PCR to determine whether treatments with DRB inhibited transcription as previously described (Miki et al., 2014). We found that 200 µg/ml of DRB partially decreased transcription in wild-type, but did not in *daf-2* mutants (Supplementary Fig. 10c). A much higher concentration of DRB (1000 µg/ml of DRB) inhibited transcription in *daf-2* mutants, whereas the same concentration of DRB killed wild-type worms. Thus, we concluded that L1 larval worms were not suitable for our mRNA half-life assays that require comparing *daf-2* mutants and wild-type animals. We therefore changed our strategy to examine the RNA half-lives of adult worms. We found that 500 µg/ml of DRB treatment successfully inhibited transcription in wild-type, *daf-2*, and *smg-2*(-); *daf-2*(-) mutant animals, although it seems that *daf-2* mutants are more resistant to DRB than wild-type at early time points (2 hr and 4 hr) (Supplementary Fig. 10d,e).”

References

- Arai, T., Hasegawa, M., Akiyama, H., Ikeda, K., Nonaka, T., Mori, H., Mann, D., Tsuchiya, K., Yoshida, M., Hashizume, Y., *et al.* (2006). TDP-43 is a component of ubiquitin-positive tau-negative inclusions in frontotemporal lobar degeneration and amyotrophic lateral sclerosis. *Biochemical and biophysical research communications* 351, 602-611.
- Bensaude, O. (2011). Inhibiting eukaryotic transcription: Which compound to choose? How to evaluate its activity? *Transcription* 2, 103-108.
- Chen, C.Y., Ezzeddine, N., and Shyu, A.B. (2008). Messenger RNA half-life measurements in mammalian cells. *Methods in enzymology* 448, 335-357.
- Dluzen, D.F., Noren Hooten, N., and Evans, M.K. (2016). Extracellular RNA in aging. *Wiley interdisciplinary reviews RNA*.
- Inukai, S., and Slack, F. (2013). MicroRNAs and the genetic network in aging. *Journal of molecular biology* 425, 3601-3608.
- Kour, S., and Rath, P.C. (2016). Long noncoding RNAs in aging and age-related diseases. *Ageing research reviews* 26, 1-21.
- Miki, T.S., Ruegger, S., Gaidatzis, D., Stadler, M.B., and Grosshans, H. (2014). Engineering of a conditional allele reveals multiple roles of XRN2 in *Caenorhabditis elegans* development and substrate specificity in microRNA turnover. *Nucleic acids research* 42, 4056-4067.
- Neumann, M., Sampathu, D.M., Kwong, L.K., Truax, A.C., Micsenyi, M.C., Chou, T.T., Bruce, J., Schuck, T., Grossman, M., Clark, C.M., *et al.* (2006). Ubiquitinated TDP-43 in frontotemporal lobar degeneration and amyotrophic lateral sclerosis. *Science (New York, NY)* 314, 130-133.
- Solis, G.M., and Petrascheck, M. (2011). Measuring *Caenorhabditis elegans* life span in 96 well microtiter plates. *Journal of visualized experiments : JoVE*.

Reviewer #3 (Remarks to the Author)

Son et al.
NCOMMS-16-01874B

This manuscript is certainly improved. The new data offering some insight into mechanism is good – using *smg-2*, *daf-2* mutants, that downregulation of the level of *yars-2*/tyrosyl tRNA synthetase mRNA by the enhanced NMD that typifies *daf-2* mutants contributes to their longevity.

Specific points

The title, abstract and other writing are still misleading since there is no effect of NMD on longevity except in the *daf-2* and possibly other mutants. For the authors to keep this title, they would have to enhance NMD activity and see longer life spans (which may or may not be the result). A more appropriate title would be more specific and note that the enhanced NMD that typifies long-lived *daf-2* mutants decreases the level of *yars-2*/tyrosyl tRNA synthetase mRNA, which augments lifespan in these mutants.

Discussion, page 10, line 207. As written, the meaning of “neurons were the most crucial tissues for the role of NMD in longevity conferred by *daf-2*/insulin/IGF-1 receptor mutations.

Also on page 10, line 212 “contributions of NMD to longevity in different tissues appear to be different” is an overstatement since all the authors can conclude is that the efficiency of NMD in different tissues is different as a consequence of aging.

Page 11, line 226. The authors speculate that *daf-2* mutations may decrease cytosolic Ca^{++} levels and up-regulate NMD. However, how does downregulated *yars-2*/tyrosyl tRNA synthetase mRNA by NMD augment lifespan of the *daf-2* mutants?

Page 11, line 233. “IIS” requires a definition.

Page 12, line 239. While NMD is a “protective system” that protects cells from mistakes in gene expression, this reviewer would not call the findings here part of a protective system but more a system that maintains cellular homeostasis, for which many examples exist in the literature.

Reviewer #1 (Remarks to the Author):

Reviewer #1 also asks that you highlight the differences between *smg-2* RNAi and mutations, as pointed out in your response to the comments made by reviewer #1 in the previous round, in the main text.

> We thank the reviewer for the comment. As the reviewer requested, we described the difference between *smg-2* RNAi and mutations in the main text as follows:

Results, page 5, line 99: “We found that *smg-2* mutations also shortened the longevity of mitochondrial respiration-defective *isp-1(-)* mutants (Fig. 2d), dietary restriction mimetic *eat-2(-)* mutants (Fig. 2e), and germline-deficient *glp-1(-)* mutants (Fig. 2f). **In contrast to the results with genetic mutants, *smg-2* RNAi did not decrease longevity induced by *isp-1* (Supplementary Fig. 2c), *eat-2* (Supplementary Fig. 2d), or *glp-1* (Supplementary Fig. 2e) mutations.**”

Figure 2 legends, page 32, line 728: “**Specific lifespan-decreasing effects of *smg-2* RNAi on *daf-2* mutants may be due to variability in RNAi efficiency, as we did not perform all the RNAi-based lifespan assays using various longevity mutants in the same experimental set.**”

Reviewer #3 (Remarks to the Author):

This manuscript is certainly improved. The new data offering some insight into mechanism is good – using *smg-2*, *daf-2* mutants, that downregulation of the level of *yars-2*/tyrosyl tRNA synthetase mRNA by the enhanced NMD that typifies *daf-2* mutants contributes to their longevity.

Specific points

The title, abstract and other writing are still misleading since there is no effect of NMD on longevity except in the *daf-2* and possibly other mutants. For the authors to keep this title, they would have to enhance NMD activity and see longer life spans (which may or may not be the result). A more appropriate title would be more specific and note that the enhanced

NMD that typifies long-lived *daf-2* mutants decreases the level of *yars-2*/tyrosyl tRNA synthetase mRNA, which augments lifespan in these mutants.

> We agree with the reviewer's comment, and included *daf-2* in our title as follows,

Title: RNA surveillance via nonsense-mediated mRNA decay is crucial for longevity in *daf-2/insulin/IGF-1 mutant C. elegans*

Discussion, page 10, line 207. As written, the meaning of “neurons were the most crucial tissues for the role of NMD in longevity conferred by *daf-2/insulin/IGF-1* receptor mutations.

> We thank the reviewer for this comment, and toned down our findings as follows,

Discussion, page 11, line 219: “Interestingly, neurons are at least partially resistant to age-dependent declines in NMD activity, and NMD in neuron contributes to longevity conferred by *daf-2/insulin/IGF-1* receptor mutations.

Also on page 10, line 212 “contributions of NMD to longevity in different tissues appear to be different” is an overstatement since all the authors can conclude is that the efficiency of NMD in different tissues is different as a consequence of aging.

> We revised our manuscript based on this comment as follows,

Discussion, page 11, line 224: “One interesting aspect of our study is that the efficiency of NMD in different tissues during the course of aging appear to be different.”

Page 11, line 226. The authors speculate that *daf-2* mutations may decrease cytosolic Ca^{++} levels and up-regulate NMD. However, how does downregulated *yars-2*/tyrosyl tRNA synthetase mRNA by NMD augment lifespan of the *daf-2* mutants?

> We appreciated that the reviewer made this valuable comment. We now have a discussion paragraph regarding how downregulated *yars-2* may increase lifespan in *daf-2* mutants as follows,

Discussion, page 11, line 231: “Our data suggest that enhanced NMD increases lifespan in *daf-2* mutants via efficiently degrading *yars-2b.1*/tyrosyl-tRNA synthetase isoform b.1.

Although underlying mechanisms are still elusive, our findings are consistent with previous findings. Genetic inhibition of tRNA synthetases, which may decrease mRNA translation, extends *C. elegans* lifespan (Kim and Sun, 2007; Lee et al., 2003). *daf-2* mutants display reduced translation rates (Stout et al., 2013), and reduced translation confers longevity in multiple organisms (Steffen and Dillin, 2016). Thus, increased NMD activity by reduced insulin/IGF-1 signaling (IIS) may prevent abnormal and age-dependent increases in transcripts, including tRNA synthetase genes. This may in turn contribute to the longevity of *daf-2* mutants by decreasing translation. It is intriguing that reduced IIS may promote healthy aging by increasing the quality of RNA and proteins by affecting both transcription and translation.”

In addition, we completely deleted the paragraph regarding our pure speculation on the relationship among *daf-2*, Ca^{2+} and NMD, because our *yars-2b.1* data are much stronger than any Ca^{2+} discussion.

Page 11, line 233. “IIS” requires a definition.

> We thank the review for the comment, and defined IIS as follows,

Discussion, page 12, line 241: “We report that **insulin/IGF-1 signaling (IIS)**, one of the most evolutionarily conserved aging-regulatory signaling pathways, regulates RNA surveillance through NMD to influence aging.”

Page 12, line 239. While NMD is a “protective system” that protects cells from mistakes in gene expression, this reviewer would not call the findings here part of a protective system but more a system that maintains cellular homeostasis, for which many examples exist in the literature.

> We thank the reviewer for this insightful comment. Based on the comment, we changed our sentence as follows,

Discussion, page 12, line 249: “As humans live much longer than animals with

comparable sizes and metabolic rates, and NMD components are well conserved, it is likely that humans are equipped with systems that maintain cellular RNA homeostasis as well.”

Best regards,

Seung-Jae V. Lee (On behalf of all the authors)

References

Kim, Y., and Sun, H. (2007). Functional genomic approach to identify novel genes involved in the regulation of oxidative stress resistance and animal lifespan. *Aging cell* 6, 489-503.

Lee, S.S., Lee, R.Y., Fraser, A.G., Kamath, R.S., Ahringer, J., and Ruvkun, G. (2003). A systematic RNAi screen identifies a critical role for mitochondria in *C. elegans* longevity. *Nature genetics* 33, 40-48.

Steffen, K.K., and Dillin, A. (2016). A Ribosomal Perspective on Proteostasis and Aging. *Cell metabolism* 23, 1004-1012.

Stout, G.J., Stigter, E.C., Essers, P.B., Mulder, K.W., Kolkman, A., Snijders, D.S., van den Broek, N.J., Betist, M.C., Korswagen, H.C., Macinnes, A.W., *et al.* (2013). Insulin/IGF-1-mediated longevity is marked by reduced protein metabolism. *Molecular systems biology* 9, 679.